**nature** COMMUNICATIONS

# BAD regulates mammary gland morphogenesis by 4E-BP1-mediated control of localized translation in mouse and human models

John Maringa Githaka[1], Namita Tripathi[1], Raven Kirschenman[1], Namrata Patel[1], Vrajesh Pandya[1], David A. Kramer [1], Rachel Montpetit[1], Lin Fu Zhu [2], Nahum Sonenberg [3], Richard P. Fahlman [1], Nika N. Danial [4], D. Alan Underhill[5] & Ing Swie Goping [1,5✉]

Elucidation of non-canonical protein functions can identify novel tissue homeostasis pathways. Herein, we describe a role for the Bcl-2 family member BAD in postnatal mammary gland morphogenesis. In $Bad^{3SA}$ knock-in mice, where BAD cannot undergo phosphorylation at 3 key serine residues, pubertal gland development is delayed due to aberrant tubulogenesis of the ductal epithelium. Proteomic and RPPA analyses identify that BAD regulates focal adhesions and the mRNA translation repressor, 4E-BP1. These results suggest that BAD modulates localized translation that drives focal adhesion maturation and cell motility. Consistent with this, cells within $Bad^{3SA}$ organoids contain unstable protrusions with decreased compartmentalized mRNA translation and focal adhesions, and exhibit reduced cell migration and tubulogenesis. Critically, protrusion stability is rescued by 4E-BP1 depletion. Together our results confirm an unexpected role of BAD in controlling localized translation and cell migration during mammary gland development.

[1] Department of Biochemistry, University of Alberta, Edmonton, AB, Canada. [2] Department of Surgery, University of Alberta, Edmonton, AB, Canada. [3] Department of Biochemistry, McGill University, Montreal, QC, Canada. [4] Department of Cancer Biology, Dana-Farber Cancer Institute, Boston, MA, USA. [5] Department of Oncology, University of Alberta, Edmonton, AB, Canada. ✉email: igoping@ualberta.ca

The BH3-only Bcl-2-associated agonist of death protein BAD[1] is best known as an apoptosis sensitizer, but has complex roles in vivo[2–4]. It is regulated by coordinated phosphorylation of three serine (S) residues S112, 136, and 155 (75, 99, and 118 in humans) that differentially modulate activity in a tissue-specific manner[5,6]. Genetic knock-out and phospho-mutant mice have apoptotic defects in the developing lymphoid compartment[7,8], diminished glycolysis in the pancreas and liver[9,10], and altered electrical activity in the brain[11]. In the breast, BAD is a prognostic marker for survival of breast cancer patients[12], and modulates mitochondrial metabolism and sensitivity to taxane chemotherapy in vitro[13,14]. Intriguingly, BAD is differentially expressed during mammary gland development in the mouse[15], and deciphering this may shed light on pathophysiology as aberrant reactivation of these developmental pathways defines breast carcinogenesis[16–18].

The female mammary gland undergoes extensive postnatal development including ductal morphogenesis in puberty, differentiation into a milk-producing organ in pregnancy/lactation and regression to a pre-pregnant state at weaning[19]. During puberty, estrogen and growth hormone trigger the formation of bulb-shaped terminal end buds (TEBs) on the branch tips of the rudimentary epithelial anlage[20]. These TEBs consist of a multi-layered mass of semi-differentiated myoepithelial and epithelial luminal cells that invade the mammary fat pad and repeatedly bifurcate to ultimately form an arborized ductal network. Individualistic cell migration within the TEBs mediates this forward growth involving partial epithelial to mesenchymal transition (EMT) with collective leader-follower characteristics[21]. Molecular mechanisms of cell motility have been exquisitely defined through 2D in vitro studies, identifying cycles of subcellular protrusion, adhesion, and contraction facilitated by the actomyosin cytoskeleton. Cell migration in 3D is more complex involving coordinated movement of groups of cells within complex microenvironments and thus knowledge gaps still exist on the molecular aspects of 3D and in vivo cell motility[22].

Since BAD is associated with breast pathophysiology through an unknown biological mechanism, here we examine the effects of BAD and BAD phosphorylation-mutants on postnatal mammary gland development. Using a genetic mouse model wherein the three S residues are replaced by alanine ($Bad^{3SA}$)[7], we observe a pubertal stage-specific delay in the elongation of the epithelial ductal tree. Unbiased proteomic screens identify that $Bad^{3SA}$ dysregulates the mRNA translational repressor protein 4E-BP1, focal adhesion and actin binding components, which together suggest defects to the cell motility machinery. It is known that localized mRNA translation stabilizes subcellular protrusions that are required for cell migration in vitro[23–25]. Consistent with this, $Bad^{3SA}$ organoids have protrusions that are lacking in hyper-phosphorylated 4E-BP1, are deficient for protrusion-localized translation, have destabilized protrusions and show inhibited cell migration and tubule outgrowth. Depletion of 4E-BP1 rescues the BAD[3SA]-mediated protrusion defect. Thus, this work identifies BAD as a modulator of mammary gland morphogenesis and shows a regulatory link to localized translation that is critical for pubertal stage-specific cell migration.

## Results

**BAD phosphorylation regulates mammary gland morphogenesis in puberty.** We used 3 genetic engineered mouse models to explore the role of BAD in postnatal mammary gland development; knock-out $Bad^{-/-}$, knock-in $Bad^{S155A}$ and $Bad^{3SA}$ where 3SA indicates alanine substitutions at S112/136/155 of the endogenous $Bad$ allele (gene in italics and protein in all uppercase)[7,8,10]. We first characterized BAD protein and phosphoprotein levels during mammary gland development in the wild-type animal. Serines 112, 136, and 155 are coordinately phosphorylated[26] and we used either anti-PS112 or anti-PS136 antibodies that were validated for western blot or immuno-fluorescence, to monitor phosphorylated BAD (P-BAD). Total BAD protein levels are similar in pubertal, nulliparous adult and involuting glands and elevated in pregnant glands (Fig. 1a). On the other hand, P-BAD is significantly elevated by ~26-fold, ~18-fold and ~11-fold in puberty, pregnant and involuting gland respectively, relative to nulliparous adult. At the cellular level, total BAD localizes to the epithelial cells lining mature ducts and TEBs of the pubertal mammary gland and is largely undetected in the surrounding stroma (Fig. 1b). Notably, P-BAD, is locally enriched in the TEBs (Fig. 1c). These results suggest that P-BAD modulates pubertal-specific processes, likely related to its TEB-localization.

We next surveyed whether phosphomutants of BAD have altered pubertal ductal morphogenesis. The mammary glands of $Bad^{+/+}$ animals undergo substantial growth and elongation during puberty (compare pubertal onset at 4wk vs puberty at 5wk; Fig. 2a–c and Supplementary Fig. 1). $Bad^{S155A}$ and $Bad^{3SA}$ glands, however, are significantly delayed with decreased ductal extension, ductal area, primary branching, and inconsistent branching (Fig. 2a–c and Supplementary Fig. 1). Mutation of S155 alone is sufficient to delay ductal elongation, although mutation of all three serines produces the strongest phenotype. In all other stages of pre-pubertal, adult, pregnant and involuting, $Bad^{3SA}$ glands are morphologically normal (Supplementary Fig. 2). These results highlight that $Bad^{3SA}$ delays pubertal mammary gland development, which fully recovers by early adulthood. $Bad^{-/-}$ animals are not phenotypically different at any stage, indicating a dominant effect of non-P-BAD (Fig. 2a–c and Supplementary Fig. 2). Thus, pubertal specific signals induce phosphorylation of BAD and blocking this in $Bad^{3SA}$ and Bad[S155A] animals, disturbs ductal morphogenesis.

**$Bad^{3SA}$-mediated defect is epithelial cell autonomous.** Ductal morphogenesis is coordinated through complex signals between the epithelial and stromal compartments[27–30]. Since the mice are whole-body genetic models, we transplanted epithelium from one genotype into the cleared fat pads of the opposite genotype to identify the origin of the defect (Fig. 2d–f). $Bad^{+/+}$ epithelium transplanted into either $Bad^{+/+}$ or $Bad^{3SA}$ animals repopulates the recipient fat pads with equal efficiency and similar ductal expansion (Fig. 2f–h), indicating that the $Bad^{3SA}$ stroma is competent for growth. In contrast, $Bad^{3SA}$ epithelium fails to grow in $Bad^{+/+}$ recipients (Fig. 2f, g), demonstrating that the defect resides within the $Bad^{3SA}$ epithelium (Fig. 2g; $\chi^2$ $p = 0.0235$).

We directly examined epithelial effects with ex vivo 3D organotypic branching assays using primary mouse epithelial cell organoids isolated from the mouse mammary gland[21,31–34]. These primary mouse organoids were embedded in extracellular matrix (ECM) mimics of Matrigel:Collagen-I, where they form multicellular cysts similar to TEBs and recapitulate ductal morphogenesis (Fig. 3a)[21,31]. BAD is expressed and phosphorylated in these branching organoids, similar to whole glands (Supplementary Fig. 3a). Time-lapse microscopy shows that $Bad^{3SA}$ organoids produce similar proportion of organoids with branches but have significantly slower branch elongation and lower number of branches per organoid (Fig. 3b–d and Supplementary Fig. 3b, c). Thus, $Bad^{3SA}$ organoids phenocopy the ductal elongation delay and confirm the epithelial-specific defect.

To establish a genetically tractable system and also investigate conservation in human cells[35], we utilized the MCF10A non-

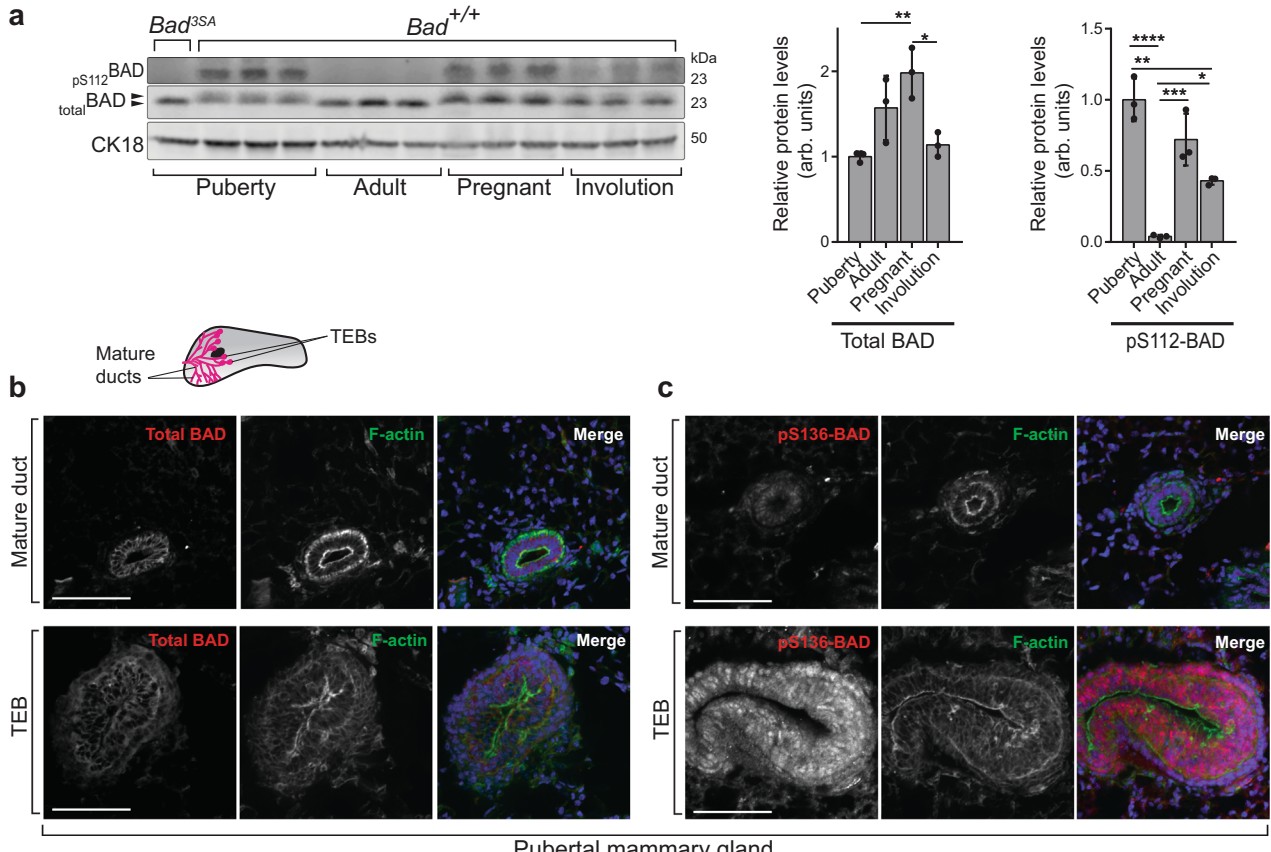

**Fig. 1 BAD is differentially phosphorylated in the developing mammary gland. a** Left: western blots of P-BAD and total BAD from $Bad^{+/+}$ mammary glands at indicated stages (Puberty at 5wk; Adult at 12wk; Pregnant; Involution). Each lane contains lysates from independent mice. Control $Bad^{3SA}$ mammary gland lysate (first lane) shows specificity of anti-pS112-BAD antibody. Black arrowheads highlight gel shift indicative of BAD phosphorylation. CK18 is used as the loading control for the epithelial compartment. Middle: quantitation of total BAD relative to CK18 loading control, shows increased BAD levels in pregnancy with similar levels in other stages. Right: quantitation of pS112-BAD relative to total BAD shows decreased phosphorylation in adult glands with highest levels in pubertal stages. Data are mean ± SD. For each stage, three independent mice were used. **b, c** Representative immunofluorescent images from $n = 3$ independent mice of $Bad^{+/+}$ pubertal mammary glands stained for **b** total BAD and **c** P-BAD with pS136BAD antibody. The pubertal epithelial tree is composed of bilayer mature ducts capped by multilayer TEBs (inset schematic). Regions representing mature ducts (upper) and TEBs (lower) within the same tissue sections are shown. Total BAD (red), pS136-BAD (red), F-actin (green) and DAPI (blue). BAD localized to the epithelial cells in all experiments. bars = 100 μm. For all p-values, ***P < 0.001, **P < 0.01, *P < 0.05. Statistical test details and exact p-values are provided in Supplementary Data 4. Source data are provided in the Source data file.

transformed breast cell line with a well-characterized model of 3D tubulogenesis[36–39]. When grown in 3D Matrigel:Collagen-I, MCF10A form multicellular cysts, where cells at the edge extend protrusions into the ECM and initiate collective cell migration to generate tubes, synonymous to mammary duct morphogenesis (Fig. 3e). We knocked-out expression of endogenous $BAD$ (human gene in all cap italics)[13] and ectopically expressed human wild-type (WT) BAD or BAD3SA (herein, referred to as WT and 3SA, respectively). All experiments were conducted alongside an additional control of parental MCF10A cells, given the variable BAD expression levels (Fig. 3f). 3SA are significantly impaired in the overall formation of tubulating cysts with slower tubule extension (Fig. 3g–i), replicating the murine ductal elongation defect. Altogether, BAD[3SA] inhibits mouse ductal elongation, with conserved effects in a 3D human cell tubulogenesis model.

**BAD[3SA] disrupts signatures related to cell motility**. To investigate the mechanism, we conducted an unbiased mass spectrometry (MS) proteomic screen comparing 5wk-$Bad^{+/+}$, 5wk-$Bad^{3SA}$ and 4wk-$Bad^{+/+}$ mammary glands. We first validated

whether the proteomic signature of $Bad^{3SA}$ glands reflected a pubertal delay. Unsupervised hierarchical correlation clustering analysis indicates that the protein profile from the 5wk-$Bad^{3SA}$ tissue clusters more closely with the 4wk rather than the age-matched 5wk-$Bad^{+/+}$ mice (Supplementary Fig. 4a, see "Methods"). Further, the overall optimal number of clusters for the protein profiles is two groups[40] (Supplementary Fig. 4b), and machine learning ensemble-based random forest analyses classifies 5wk-$Bad^{3SA}$ profiles as 4wk-$Bad^{+/+}$ (Supplementary Fig. 4c, see "Methods"). Finally, Gene Ontology (GO) enrichment analysis produces strikingly similar molecular function and cellular function profiles between 5wk-$Bad^{+/+}$ vs 4wk-$Bad^{+/+}$, and 5wk-$Bad^{+/+}$ vs 5wk-$Bad^{3SA}$ (Supplementary Fig. 4d and Supplementary Data 1). Thus, the proteomic profile of $Bad^{3SA}$ glands confirms the pubertal delay phenotype (see Fig. 2).

To identify cellular processes that are altered by BAD[3SA], we analyzed the top GO hits. There are significant changes in the molecular function, "actin binding," and the cellular component, "focal adhesion", signifying that $Bad^{3SA}$ disrupts cell migration processes (Supplementary Fig. 4d). Actin and focal adhesions are required for nascent protrusions to transition into stable protrusions that then connect to the ECM and facilitate cell

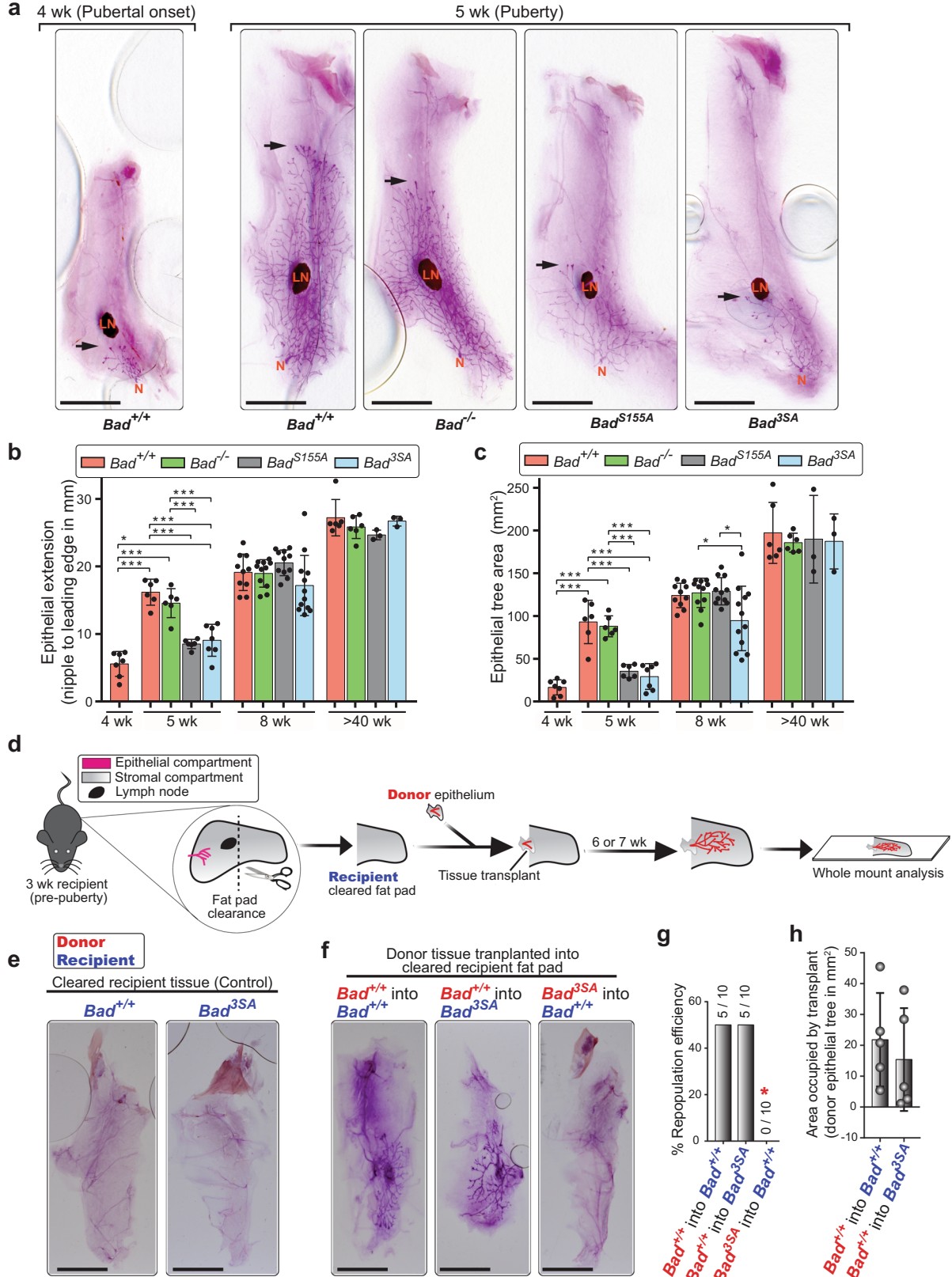

motility[41],[42]. To test whether 3SA disrupts this process, we analyzed cell protrusion dynamics at day 3 of the 3D tubulogenesis assay (see Supplementary Movie 1). Figure 4 shows how cells at the edges of the multicellular cyst extend and contract protrusions into the ECM (Fig. 4a; green arrowheads). Upon focal adhesion maturation, these protrusions stabilize and facilitate collective cell migration to form tubules (Fig. 4a; red arrowheads). MCF10A WT and 3SA-expressing cysts are equally proficient at forming nascent protrusions (Fig. 4b), however, the 3SA protrusions are significantly less stable (~3× decreased lifetime) and shorter (~1.7× decreased length), revealing defects in adhesion stabilization (Fig. 4c). Nascent adhesions require

**Fig. 2 *Bad^3SA* shows defective mammary gland development in puberty that is derived from the epithelial compartment. a** Representative carmine-stained mammary gland whole mounts (WMs) showing pubertal morphogenesis of *Bad* mutants. Shown are the #4 mammary glands of 5wk mice of *Bad^+/+* (n = 6), *Bad^−/−* (n = 6), *Bad^S155A* (n = 6), and *Bad^3SA* (n = 7). For comparison, a control *Bad^+/+* gland at 4wk (left, n = 7) representing early pubertal onset is included. The epithelial tree (dark pink branched structure) is embedded within the mammary fat pad (surrounding light pink stroma). N, nipple; LN, lymph node. Arrows highlight front of migrating epithelial tree. Scale bars = 5 mm. **b, c** Quantitation of epithelial tree morphological features for *Bad^+/+*, *Bad^−/−* *Bad^S155A*, and *Bad^3SA* at different developmental times (5wk, 8wk, >40wk). **b**) Quantitation of epithelial tree extension shows delay in *Bad^S155A* (gray) and *Bad^3SA* (blue) at 5wk. **c** Quantitation of epithelial tree area shows decreased area in *Bad^S155A* (gray) at 5wk and *Bad^3SA* (blue) at 5wk and 8wk. Data are mean ± SD. Number of animals: For pubertal onset (4wk), *Bad^+/+* n = 7. For puberty stage (5wk), towards end of puberty (8wk) and aged virgins (>40wk), *Bad^+/+* n = 6, 10, 6; *Bad^−/−* n = 6, 11, 6; *Bad^S155A* n = 6, 11, 3; and *Bad^3SA* n = 7, 12, 3, respectively. **d** Transplant assay schematic showing fat pad clearance in recipient mouse and subsequent donor tissue transplant. Donor tissue growth was assessed 6 or 7 weeks post transplant surgery. **e** Representative carmine-stained WMs of negative control "recipient" mice (n = 5 mice per genotype) that did not receive transplant shows the efficiency of fat pad clearance. Scale bars = 5 mm. **f** Representative carmine-stained WMs for experimental transplant series (n = 5 mice per transplant condition). Red text represents the donor tissue genotype and blue text represents the recipient genotype. Scale bars = 5 mm. **g** Percentage of cleared fat pads with successful transplant repopulation show failure of *Bad^3SA* epithelium to repopulate fat pad (red asterisk). The ratio of successful repopulation per total transplants is shown on top of the bars. n = 10 for each group. Computed Chi-square ($\chi^2$) p-value = 0.0235. **h** Quantitation of area occupied by transplanted *Bad^+/+* epithelium in either *Bad^+/+* or *Bad^3SA* fat pads show similar repopulation areas. Data are mean ± SD. n = 5 for each group. For all p-values, ***P < 0.001, **P < 0.01, *P < 0.05. Statistical test details and exact p-values are provided in Supplementary Data 4. Source data are provided in the Source data file.

actin polymerization to mature into stable focal adhesions that provide traction for cell migration[41]. Indeed 3SA cell protrusions are deficient for both filamentous actin and the adhesion adapter protein paxillin (Fig. 4d, e). In contrast, WT protrusions show strong F-actin staining and punctate localization of paxillin, indicative of mature focal adhesions that mark protrusion stabilization (Fig. 4d, e). Intriguingly, 3SA does not decrease paxillin and F-actin staining in interior body cells within the multicellular cyst (Fig. 4e), indicating that 3SA specifically diminishes protein accumulation locally.

**BAD^3SA defect does not depend on apoptosis.** BAD^3SA has been shown to induce apoptosis[7], so we examined whether apoptotic signaling mediates protrusion destabilization. As expected, cleaved caspase 3 is detected in the TEBs of the pubertal mammary gland[20] but is not significantly increased in *Bad^3SA* (Supplementary Fig. 5a). Additionally, blocking caspase activity with the pan caspase inhibitor zVAD-fmk does not rescue protrusion defects in BAD^3SA tubulogenesis assays (Supplementary Fig. 5b, c; Supplementary Movie 2). BAD stimulates apoptosis by binding to anti-apoptotic Bcl-XL[43]. Although BAD^3SA binds strongly to Bcl-XL in the 3D culture system, disrupting this interaction with the BH3-mimetic ABT-737 (Supplementary Fig. 5d) does not alter the ability of BAD^3SA to inhibit protrusion stability (Supplementary Fig. 5b). Thus, the mechanism whereby BAD^3SA inhibits ductal elongation does not require Bcl-XL interaction or caspase activity and is independent of apoptosis.

**BAD^3SA defect does not alter epithelial cell lineage.** We next tested if BAD^3SA alters epithelial cell lineage or the proportion of stem-like cells (Supplementary Fig. 6). To examine epithelial cell lineage, primary mouse mammary epithelial cells were stained for the surface markers CD24 versus CD49f[44]. There is no significant difference in levels or proportion of luminal or basal epithelial subtypes. We next examined the epithelial stem/progenitor pools with the markers EpCAM versus CD49f to identify the Mammary Repopulating Unit (MRU) stem cells[45]. MRUs can generate an entire functional mammary gland from a single cell[46]. There is no difference in the MRU between the genotypes. Therefore, the *Bad^3SA* defect likely manifests downstream of epithelial lineage commitment.

**BAD^3SA impedes 4E-BP1 hyperphosphorylation.** To gain molecular insight into novel pathway(s) deregulated by *Bad^3SA*,

we performed a reverse phase protein array (RPPA) antibody-based screen. By quantitating protein/phosphoprotein levels of key regulators of cancer and developmental pathways[47,48], RPPA can identify signal transduction differences with higher sensitivity than our MS-based screen[49]. BAD is phosphorylated in both 4wk and 5wk *Bad^+/+* glands but not in *Bad^3SA*, serving as an excellent internal control for RPPA specificity (Fig. 5a, b). Similar to the MS results, unsupervised hierarchical correlation clustering analysis and random forest classification shows a closer alignment between 5wk-*Bad^3SA* and 4wk-*Bad^+/+* glands rather than to the age-matched *Bad^+/+* confirming *Bad^3SA* imparts a developmental delay (Supplementary Fig. 7a, b). Analysis of variance and Tukey-Kramer multiple comparison post-hoc analysis was used to identify significant differences between all groups (Supplementary Data 2). Pathway analysis of significantly different hits (Supplementary Data 2) identifies "Focal Adhesion-PI3K-Akt-mTOR" as the most enriched expressed signaling pathway (Supplementary. Data 3, Supplementary Fig. 7c), confirming that *Bad^3SA* disrupts focal adhesion processes. Of the 8 most significantly different protein/phosphoproteins between age-matched *Bad^+/+* and *Bad^3SA* (p-value < 0.01), the top 3 hits, mTOR, 4E-BP1, and eIF4E (Fig. 5a, b), are all markers of the top-ranked Focal Adhesion-PI3K-Akt-mTOR pathway. Since mTOR, 4E-BP1, and eIF4E are modulators of mRNA translation, these results implicate an unexpected role for BAD in this pathway.

The mTORC1 kinase complex stimulates the translation of capped mRNAs in part by phosphorylating and inhibiting the translational repressor, 4E-BP1[50–52]. In its hypophosphorylated state, 4E-BP1 binds the cap-binding protein eIF4E blocking subsequent recruitment of eIF4G, which is required to form the initiation complex eIF4F[52]. mTORC1 sequentially phosphorylates 4E-BP1 at T70 and S65 to generate the hyperphosphorylated state, causing the release of eIF4E to allow eIF4F complex assembly and translation initiation[50–52]. We verified RRPA hits on independently collected mammary gland tissue lysates and 3D cultures of murine organoids or human MCF10A cells. mTOR differences are not consistent in all systems analyzed (Fig. 5b–e). Dysregulation of the mTORC1 downstream target, 4E-BP1 however, is confirmed in all models (Fig. 5b–e). 4E-BP1 is significantly hyperphosphorylated in normal 5wk pubertal glands compared to both 4wk and 5wk-*Bad^3SA* (Fig. 5d). This suggests that cap-dependent translation naturally increases during pubertal development and this is blocked by *Bad^3SA*. Lysates from primary *Bad^3SA* expressing mouse organoids and MCF10A

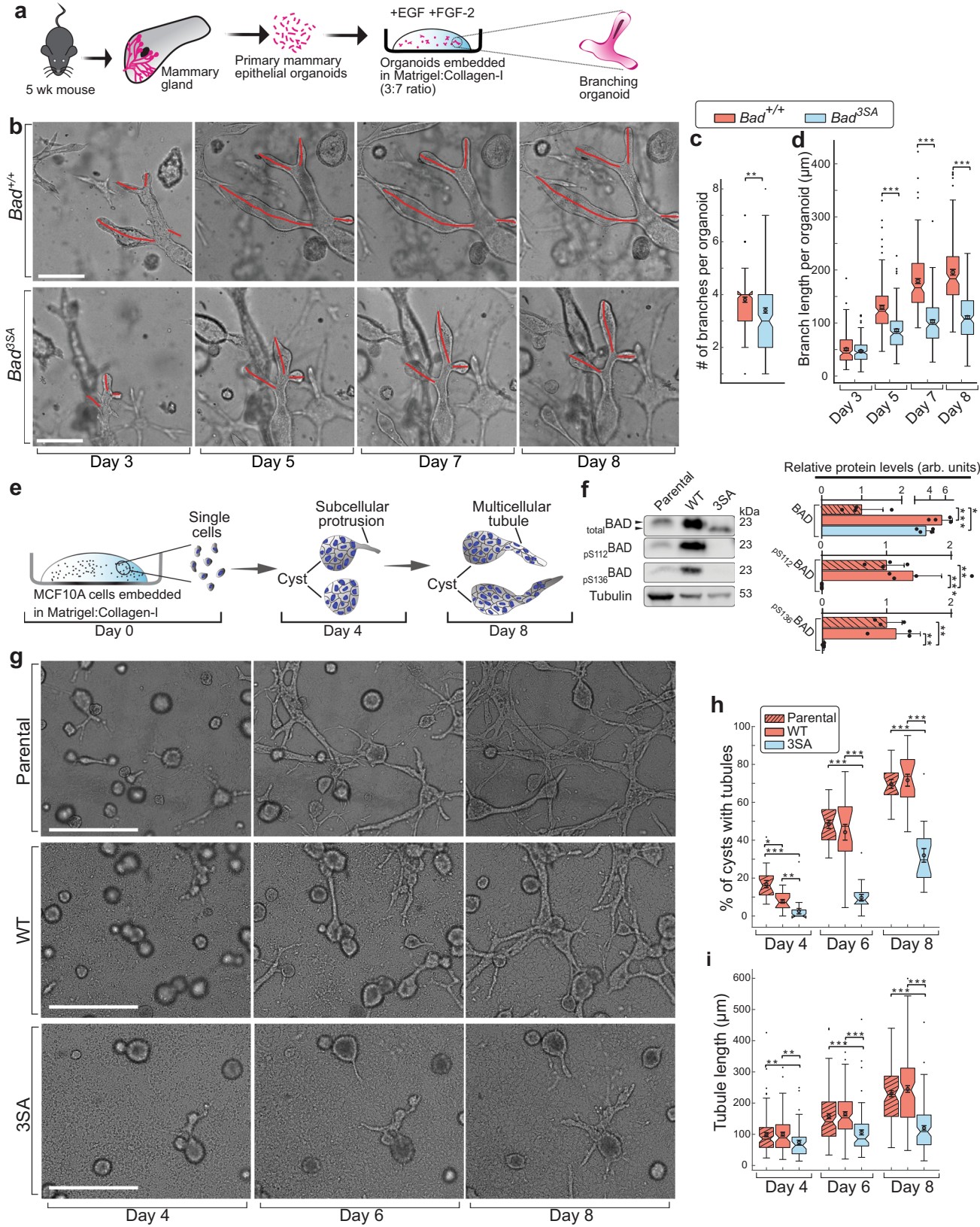

tubules also have decreased levels of hyperphosphorylated 4E-BP1 (4E-BP1_p65) (Fig. 5e and Supplementary Fig. 7g). To confirm non-significant outcomes from the RPPA, there are no differences in total and active AKT and ERK1/2 kinases, which both contribute to mTORC1 activation and mammary gland morphogenesis[53,54] (Supplementary Fig. 7d–g). Altogether, mammary epithelial cells expressing BAD[3SA] have elevated hypophosphorylated 4E-BP1 species, predicting translational repression via inhibitory complex formation between hypophosphorylated 4E-BP1 and eIF4E.

**Fig. 3 3D tissue organoid branching and MCF10A tubulogenesis assays show non-phosphorylatable BAD imparts defects. a** Schematic for 3D mouse organoid branching assay. EGF, epidermal growth factor; FGF-2, basic fibroblast growth factor. **b** Representative bright-field time-lapse images for $Bad^{+/+}$ (upper) and $Bad^{3SA}$ (lower) organoid branching assays from three independent experiments, 2 replicates per experiment. $Bad^{+/+}$ $n = 150$ organoids, $Bad^{3SA}$ $n = 185$ organoids. Red lines highlight individual branches used for quantitative measurements in (**c, d**). Scale bars = 200 μm. **c** Quantitation of number of branches per organoid. Data are represented as Tukey boxplots between $Bad^{+/+}$ (orange) and $Bad^{3SA}$ (blue) organoids. **d** Quantitation of mean branch length per organoid. Data are represented as Tukey boxplots and show decreased branch length in $Bad^{3SA}$ (blue) organoids. Three independent experiments, 2 replicates per experiment. $Bad^{+/+}$ $n = 150$ organoids, $Bad^{3SA}$ $n = 185$ organoids. **e** Schematic for 3D MCF10A tubulogenesis assay. **f** Left: western blots showing total BAD, pS112BAD, and pS136BAD in normal MCF10A (Parental) and WT and 3SA cells. Lysates were prepared from 3D tubulogenesis assays (day 7). Black arrowheads highlight gel shift indicative of BAD phosphorylation. Right: Quantitation of western blots. Data are mean ± SD of four independent experiments. **g** Representative bright-field time-lapse images for 3D MCF10A tubulogenesis assays of Parental (upper), WT (middle), and 3SA (lower) from three independent experiments. Fields of views for Parental $n = 20$, WT $n = 21$, 3SA $n = 20$. Scale bars = 500 μm. **h** Percentage of cysts with branches (i.e., undergoing tubulogenesis). Tukey boxplots show decreased proportion of branching cysts in 3SA (blue). Fields of views analyzed, Parental $n = 20$, WT $n = 21$, 3SA $n = 20$. **i** Quantitation of mean tubule length. Tukey boxplots show decreased tubule length in 3SA (blue). Three independent experiments, 2 replicates per experiment. Number of tubules analyzed, Parental $n = 90$, WT $n = 103$, 3SA $n = 104$. For Tukey boxplots, constriction indicates median, notch indicates 95% confidence interval, box edges are 25th and 75th percentiles, whiskers show extreme data points, 'outliers' plotted as black dots, overlaid circle, and error bars are the mean ± SEM. For all $p$-values, ***$P < 0.001$, **$P < 0.01$, *$P < 0.05$. Statistical test details and exact $p$-values are provided in Supplementary Data 4. Source data are provided in the Source data file.

**BAD$^{3SA}$ dysregulates 4E-BP1 interactions and localization.** Hypophosphorylated 4E-BP1 inhibits translation by binding eIF4E and occluding the recruitment of eIF4G, which is normally required for subsequent eIF4F formation and translation initiation[50–52]. To examine whether 3SA disrupts normal eIF4E protein interactions, we used a m$^7$GTP-cap pull-down assay to isolate cap-bound eIF4E. Indeed, there is increased 4E-BP1 associated with eIF4E, with corresponding decreased association of eIF4G to eIF4E in 3SA-expressing cells compared to parental MCF10A and WT (Fig. 6a). Since 3SA induces protrusion-specific defects, we next assessed whether 4E-BP1 or eIF4E are differentially localized to protrusions. The body cells within the multicellular cysts and subcellular protrusions show similar levels of total eIF4E and 4E-BP1 in both WT and 3SA (Fig. 6b; Supplementary Fig. 8a). Hyperphosphorylated 4E-BP1 (p65_4E-BP1), however, is differentially expressed. It is similarly expressed in the body cells between the two genotypes, yet intriguingly, hyperphosphorylated 4E-BP1 is significantly reduced in 3SA protrusions (~2-fold decrease). Thus, 3SA protrusions are enriched for hypophosphorylated 4E-BP1 that is bound to eIF4E and prevents recruitment of eIF4G. Altogether these results suggest that 3SA inhibits mRNA translation locally within protrusions.

**BAD$^{3SA}$ inhibits localized mRNA translation with reduced protrusion stability and cell motility.** Cell migration requires compartmentalized assembly of regulatory signaling molecules and cytoskeletal/adhesion proteins at the sites of cellular protrusions. Much of this spatial organization is driven by localized mRNA translation[23,25,55,56]. Given that we observed protrusion-specific depletion of F-actin and paxillin (Fig. 4d, e), we hypothesized that 3SA disrupted localized translation required for motility. To test whether 3SA indeed diminishes translation, we evaluated protein synthesis in 3D tubulogenesis assays. Using the surface sensing of translation (SUnSET) assay[57], puromycin is incorporated into growing polypeptide chains generating truncated puromycinylated peptides[58] that are visualized with anti-puromycin antibodies[23,57] (Supplementary Figs. 8b, c and 9). $Bad^{+/+}$ 3D primary mouse organoid cultures have a significant ~2-fold increased puromycinylated peptide staining in areas of branch elongation relative to cells within the organoid body (Supplementary Fig. 8d). This localized protein synthesis is reduced in $Bad^{3SA}$ organoid branches. Parental and WT MCF10A tubules similarly have elevated translation in areas of tubule elongation that is significantly decreased in 3SA protrusions (Fig. 6c; Supplementary Fig. 8e). In support of a regional effect, global translation is unaffected by 3SA (Supplementary Fig. 8b, c).

Further, a bicistronic fluorescent reporter assay shows that 3SA reduced cap-dependent and -independent translation in protrusions (Supplementary Fig. 8f). Thus, mRNA translation is enriched in localized areas of ductal/tubule elongation in both primary mouse and human cell line 3D organoid analyses, and this is inhibited by $Bad^{3SA}$.

**4E-BP1 inhibits protrusion stabilization.** If 3SA-dependent motility defects are indeed due to translational inhibition, then downregulation of 4E-BP1 should rescue the defect. To test this, we generated stable 4E-BP1 knockdowns in the WT and 3SA MCF10A cell lines (Fig. 7a) and conducted 3D tubulogenesis time-lapse analyses. 3SA nascent subcellular protrusions have a significantly shorter lifetime than WT cells, and this is completely rescued with 4E-BP1 knockdown (Fig. 7b, c and Supplementary Movie 3), demonstrating that 3SA destabilizes protrusions via 4E-BP1. Interestingly, 4E-BP1 knockdown increases protrusion stability in both 3SA and WT cells, such that nearly all cysts have stable protrusions by the end of the time course (Fig. 7f). Interestingly, these stable protrusions, are incompetent for elongation and do not form tubes (Fig. 7c–e, g and Supplementary Movie 3). This suggests that 4E-BP1 destabilizes protrusions not only in 3SA-expressing cells, but also in normal control nascent protrusions. Therefore 4E-BP1 blocks the transition from nascent to stable protrusions. 3SA inhibits protrusion stabilization by facilitating 4E-BP1 hypophosphorylation, thus supporting the ability of 4E-BP1 to block translation and protrusion elongation.

## Discussion

Our data identify unexpected roles for BAD phosphorylation and localized mRNA translation in mammary gland development. We propose a working model whereby in puberty, ductal epithelial cells stimulate compartmentalized translation via localized phosphorylation/inactivation of the translational inhibitor 4E-BP1. Non-P-BAD (BAD$^{3SA}$) blocks this hyperphosphorylation of 4E-BP1 such that hypophosphorylated 4E-BP1 binds and represses eIF4E-mediated translation. This thereby represses localized translation required for focal adhesion maturation, cell protrusion stability, cell motility, and ultimately mammary gland morphogenesis (Fig. 8).

An intriguing observation is that while non-P-BAD knock-in mice ($Bad^{3SA}$) have delayed mammary gland pubertal development, the Bad knock-out animals are phenotypically normal. In fact, even though Bad expression is upregulated in the mammary gland during late pregnancy, lactation, and involution, our study

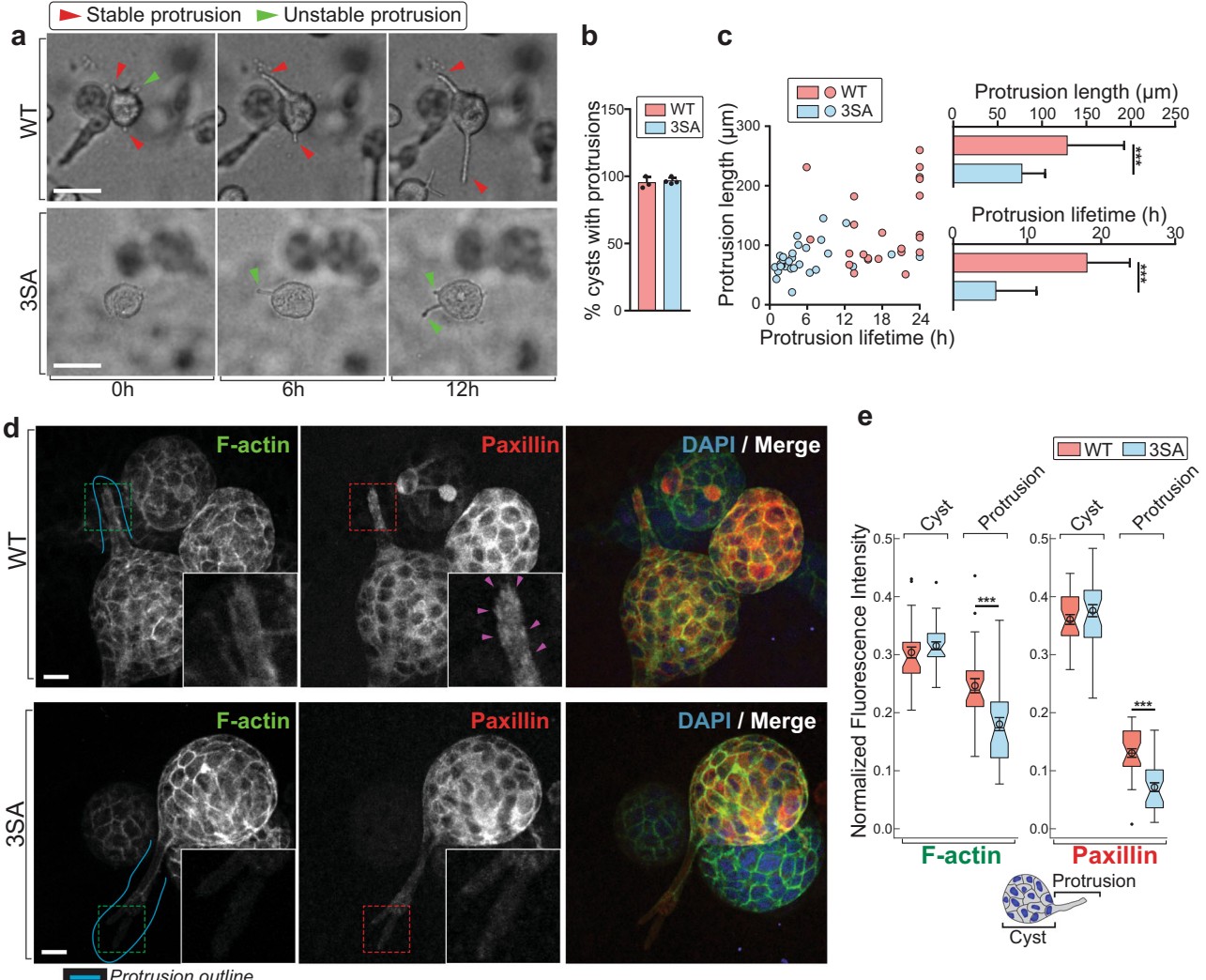

**Fig. 4 3D MCF10A cysts from 3SA have unstable protrusions. a** Representative bright-field time-lapse images of WT (upper) and 3SA (lower) MCF10 cysts in early 3D tubulogenesis assays, from three independent experiments. WT $n = 23$, 3SA $n = 29$ cysts (for complete time-lapse images, see Supplementary Movie 1). Individual subcellular protrusions either lengthen over time (red arrowheads) or are unstable and retract (green arrowheads). Scale bars = 100 μm. **b** Percentage of cell cysts that produced nascent protrusions (% cysts with protrusions) was not different between WT and 3SA. 3 independent experiments. **c** The length and lifetime of individual protrusions are shown as a scatter plot comparing WT (orange dots) and 3SA (blue dots). Quantitation shows decreased protrusion length (upper) and decreased protrusion lifetime (lower) in 3SA (blue). WT $n = 23$, 3SA $n = 29$, from three independent experiments. **d** Representative immunofluorescence images of cysts from WT (upper) and 3SA (lower) early tubulogenesis assays (day 4) stained with phalloidin (F-actin, green), anti-paxillin (focal adhesion marker, red) and DAPI (blue). DAPI highlights the nuclei in the multicellular cysts, and absence or presence of nuclei in the protrusions. The teal line is placed to outline subcellular protrusions subject to subsequent quantification analyses. Insets represent magnified areas indicated by dashed line boxes. Magenta arrowheads in the paxillin channel point to focal adhesion plaques. Scale bars = 20 μm. WT $n = 30$, 3SA $n = 33$, from three independent experiments. **e** Tukey boxplots show quantitation for normalized F-actin (left) or paxillin (right) fluorescence intensity in either the cells within the cyst (Cyst) or subcellular protrusions (Protrusion). Below is a schematic of the multicellular cyst (Cyst) with a subcellular protrusion (Protrusion). WT and 3SA show similar F-actin and paxillin staining in the cells within the cyst (Cyst). 3SA (blue) shows decreased F-actin (left) and paxillin (right) staining in the protrusion (Protrusion). WT $n = 30$, 3SA $n = 33$, from three independent experiments. For Tukey boxplots, constriction indicates median, notch indicates 95% confidence interval, box edges are 25th and 75th percentiles, whiskers show extreme data points, "outliers" plotted as black dots, overlaid circle, and error bars are the mean ± SEM. For all $p$-values, ***$P < 0.001$, **$P < 0.01$, *$P < 0.05$. Statistical test details and exact $p$-values are provided in Supplementary Data 4. Source data are provided in the Source data file.

and others show that the knock-out mammary gland is phenotypically normal[15,59]. Our study, therefore, indicates that P-BAD is relatively inert, whereas forced expression of non-P-BAD ($Bad^{3SA}$) has a dominant effect. This likely has in vivo relevance because $Bad^{3SA}$ only affected pubertal development, a stage when wild-type BAD is normally phosphorylated. Wild-type P-BAD localizes to the TEBs, which are structures that appear in puberty and regress in adulthood[60]. As such BAD is not phosphorylated in the post pubertal mammary gland, when cell migration ceases.

Thus, BAD is normally phosphorylated in puberty and passively permits cell migration in the TEBs. Its non-phosphorylated counterpart in $Bad^{3SA}$ actively blocks migration.

$Bad^{3SA}$ delays pubertal development but does not fully block ductal morphogenesis. This suggests that cells utilize compensatory mechanisms to achieve motility, as described in a 'multi-parameter tuning model'[61]. Indeed, other genetic studies show similar transient pubertal delay and interestingly, those targeted genes fall within the signaling pathway affected by $Bad^{3SA}$. For

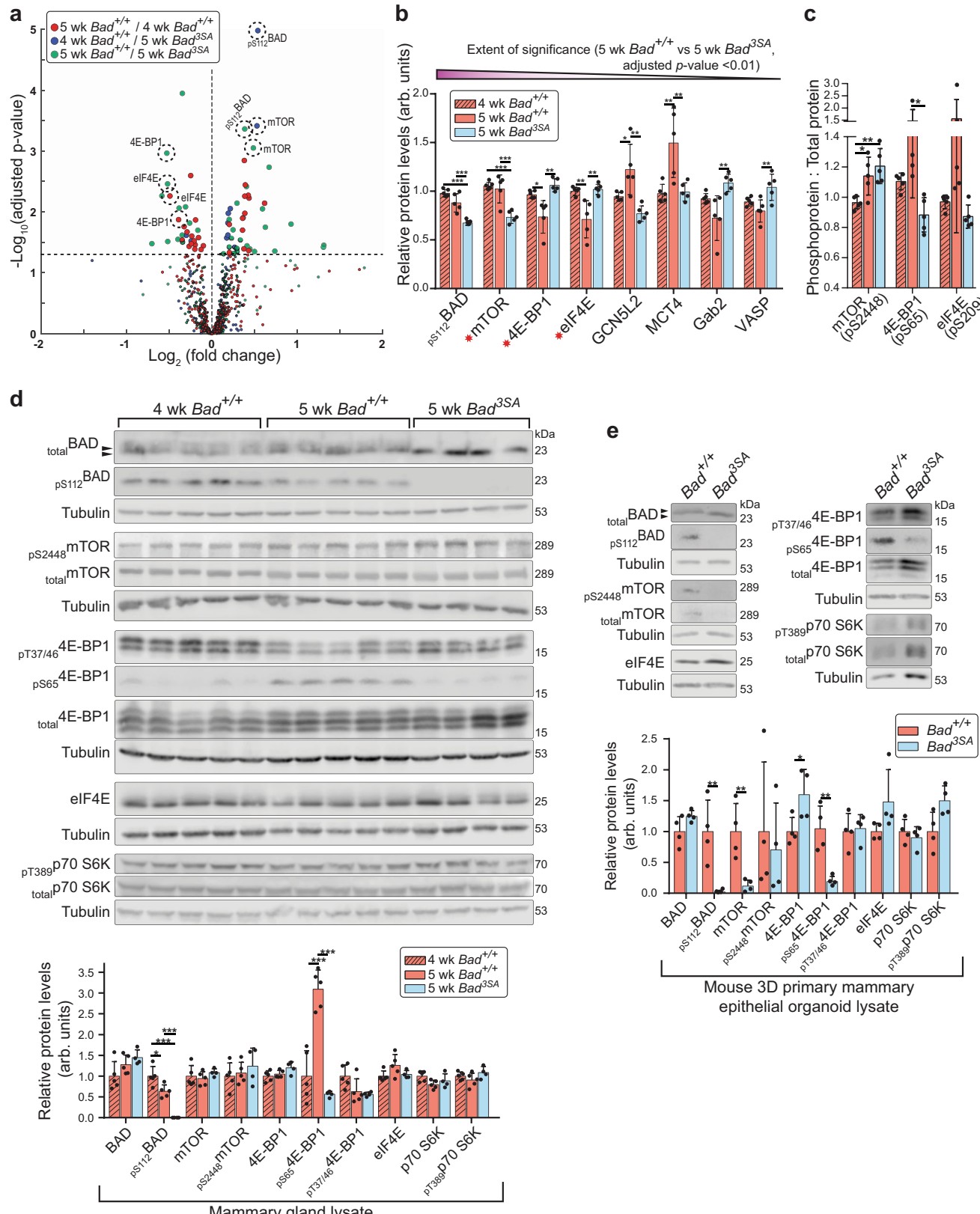

example, a pubertal delay phenotype occurs with in vivo depletion studies of mTORC1[62], which is consistent with our model where *Bad^3SA* disrupts a key mTORC1 target, 4E-BP1. Ductal pubertal delay has also been reported in FGFR-null mammary glands[63], suggesting that BAD phosphorylation is downstream of FGFR signaling. This is consistent with our study, as FGF induces

tubulogenesis in primary mouse 3D organoid cultures, and tubulogenesis is blocked by *Bad^3SA*. Furthermore, FGFR signaling responds to the pubertal hormones estrogen and progesterone[64], providing an explanation for the pubertal-specific effect of *Bad^3SA*. Additionally, stromal depletion of Sharpin[65] also phenocopies *Bad^3SA*. In this case, loss of Sharpin decreases ECM

**Fig. 5 RPPA screen identifies molecular markers that are altered in Bad^{3SA}. a** Volcano plot of mammary gland protein and phosphoprotein differences between either different pubertal stages (4wk versus 5wk) or different genotypes ($Bad^{+/+}$ versus $Bad^{3SA}$). X-axis is the fold change between groups ($Log_2$ transformed) and the Y-axis is the $-Log_{10}$ adjusted p-value from Tukey-Kramer post hoc tests between groups (see Supplementary Data 2). The dotted horizontal line marks the significance alpha (p-value = 0.05). Dotted circles indicate the top 8 significant hits that represent 4 individual proteins or their phosphoproteins as shown in (**b**). **b** Total/phosphoprotein hits were ranked by order of significance (left to right on bar graph), based on differences between 5wk $Bad^{+/+}$ (orange) and $Bad^{3SA}$ (blue) (p-value < 0.01, only RPPA validated antibodies). $Bad^{+/+}$ 4wk n = 5, $Bad^{+/+}$ 5wk n = 5, $Bad^{3SA}$ 5wk n = 5 independent mice. Data are mean ± SD. **c** Proportion of phosphorylated species for proteins asterisked in (**b**). $Bad^{+/+}$ 4wk n = 5, $Bad^{+/+}$ 5wk n = 5, $Bad^{3SA}$ 5wk n = 5 independent mice. Data are mean ± SD. **d** Top: western blots of mammary gland lysates for the top differentially expressed total/phosphoproteins identified by RPPA. Validation lysates were generated from mammary glands independent from RPPA samples. Each lane contains lysates from independent mice ($Bad^{+/+}$ 4wk n = 5, $Bad^{+/+}$ 5wk n = 5, $Bad^{3SA}$ 5wk n = 4). Black arrowheads highlight gel shift indicative of BAD phosphorylation. Bottom: quantitation of relative protein levels validated that pS112BAD and pS65-4E-BP1 are significantly decreased in $Bad^{3SA}$ (blue). Data are mean ± SD. **e** Top: western blots from mouse 3D branching organoid lysates probing top total/phosphoprotein hits identified in (**b**). Lysates were made from day 6 of organoid assays. Bottom: quantitation of relative protein levels show $Bad^{3SA}$ (blue) had decreased pS112BAD, decreased pS65-4E-BP1 and increased levels of total 4E-BP1. Data are mean ± SD of four independent experiments. For all p-values, ***P < 0.001, **P < 0.01, *P < 0.05. Statistical test details and exact p-values are provided in Supplementary Data 4. Source data are provided in the Source data file.

collagen stiffness, diminishing integrin signaling. We speculate that by reducing the translation of focal adhesion components such as paxillin and actin, BAD^{3SA} would similarly impair integrin function. Clearly, our data integrates a novel BAD/4E-BP1 axis into known molecular pathways of mammary gland morphogenesis.

We demonstrate that non-P-BAD inhibits pubertal development by interfering with mRNA translation. Translation is controlled by the kinase complex mTORC1 and mTORC1 loss-of-function induces similar transient pubertal mammary gland delay[62]. We, therefore, assessed whether $Bad^{3SA}$ inhibits mTOR. Although $Bad^{3SA}$ primary 3D organoids decrease both total and P_S2448 mTOR levels (Fig. 5e), these differences are not recapitulated in either whole mammary glands or MCF10A 3D tubules (Fig. 5d, Supplementary Fig. 7g). Additionally, the mTORC1 downstream target p70-S6K is not differentially phosphorylated in any of the experimental models (Fig. 5d, e, Supplementary Fig. 7g and Supplementary Data 2), suggesting mTORC1 is not the target of the $Bad^{3SA}$ defect. On the other hand, $Bad^{3SA}$ consistently disrupts regulatory phosphorylation of the translational inhibitor, 4E-BP1. 4E-BP1 is also classically regulated by mTORC1 via sequential phosphorylation of T37, T46, and S65[50,66,67]. Notably, phosphorylation of 4E-BP1 (T37/46) is not different between the genotypes, again ruling out a direct role of mTORC1. Instead, $Bad^{3SA}$ specifically blocks 4E-BP1 only at its hyperphosphorylation site (S65). Taken together, these results suggest that $Bad^{3SA}$ regulates 4E-BP1 downstream of, or independent of, mTORC1. In a potentially similar scenario, RhoE regulates actin and focal adhesion assembly of NIH3T3 cells by inhibiting phosphorylation of 4E-BP1 on S65, independent of mTOR[68]. Thus, while the mTOR/4E-BP1 axis is well established, alternative 4E-BP1 kinases and phosphatases are known[66] and may be regulated by $Bad^{3SA}$. Candidate P-S65_4E-BP1 kinases include GSK3β, ERK1/2, PIM2, p38MAPK, CDK1, and CDK2[66,69,70]. While Bad^{3SA} mammary gland lysates show no differences in phosphorylation of regulatory sites in 3 of these kinases (GSK3β, CDK1, or ERK1/2; Supplementary Data 2), the contribution of other unexplored kinases or phosphatases is unknown at this point. Indeed, this might explain why 4E-BP1 phosphorylation is unaffected by mTOR inhibitors in some cancer cells[71–73]. The mechanism by which BAD^{3SA} inhibits 4E-BP1 hyperphosphorylation is speculative at this point. Non-P-BAD is localized to mitochondria and enhances mitochondrial metabolism[13,74] and GO analysis of mammary glands from this study, identify mitochondria and ATP synthase activity as differentially represented in $Bad^{3SA}$ tissue (Supplementary Fig. 4d). Altered mitochondrial metabolism may additionally influence amino acid metabolism that is critical for protein synthesis.

Intriguingly, leucine depletion decreases mRNA translation in association with hypophosphorylated 4E-BP1[75]. Thus, further investigations will be needed to elucidate the molecular interactions whereby non-P BAD modulates mitochondrial metabolism, and whether this alters 4EBP-1 phosphorylation in the migrating pubertal mammary epithelial cell.

Bad3SA disrupts pubertal mammary gland development and alters cell migration. While stem cells are critical for mammary gland morphogenesis, $Bad^{3SA}$ does not appear to affect the stem/progenitor pools. $Bad^{3SA}$ does not alter MRUs, which are capable of regenerating a functional mammary tree from a single stem cell[46]. These MRUs serve as a source of differentiating luminal and myoepithelial cells and in line with this, the $Bad^{3SA}$ mammary gland also has normal cell lineage proportions. Thus, $Bad^{3SA}$ alters a morphogenetic process that is downstream of epithelial lineage commitment. Since epithelial cell motility is also critical for ductal elongation[76], $Bad^{3SA}$-mediated defects in cell motility are likely the cause of the developmental delay. During the process of cell motility, local translation efficiently accumulates newly synthesized proteins at leading protrusions[77,78]. Inhibiting this local translation destabilizes nascent protrusions and diminishes cell motility in 2D assays[23–25,56,79]. Local translation products that are critical for cell migration include modulators of actin polymerization and adhesion complexes[80,81]. In fact, local enrichment of actin and actin-related proteins is required for collective elongation of mammary epithelial tubes[82]. In line with this our MS proteomic screen identified pubertal-induced differential protein expression in actin binding and focal adhesion proteins, and we validated that BAD^{3SA} protrusions are diminished for F-actin and paxillin. In particular, β-actin mRNA is locally translated near focal adhesions[83] and interestingly, it is the newly synthesized β-actin that is preferentially used in focal adhesion maturation[56]. While global β-actin is less sensitive to cap-dependent translation perturbation[84–88], this has recently been demonstrated to be cell type and stress dependent[89–91]. Indeed, during axon guidance, localized β-actin translation is associated with localized 4E-BP1 hyperphosphorylation[92]. Disruption of β-actin localized translation decreases cell protrusion stability, reduces focal adhesion maturation, and inhibits cell motility[56,93], which similarly describe the BAD^{3SA} phenotype, suggesting that mammary epithelial cell migration requires compartmentalized protein synthesis of β-actin. Of course, β-actin is not the only locally translated mRNA that contributes to cell motility. Global screens and mRNA/protein quantitation of cell protrusions in MDA-MB-231 breast carcinoma cells established that many actin-associated protein mRNAs are translated at protrusions including Arp2/3 and alpha-actinin among others[23]. These proteins are also depleted in $Bad^{3SA}$ mammary

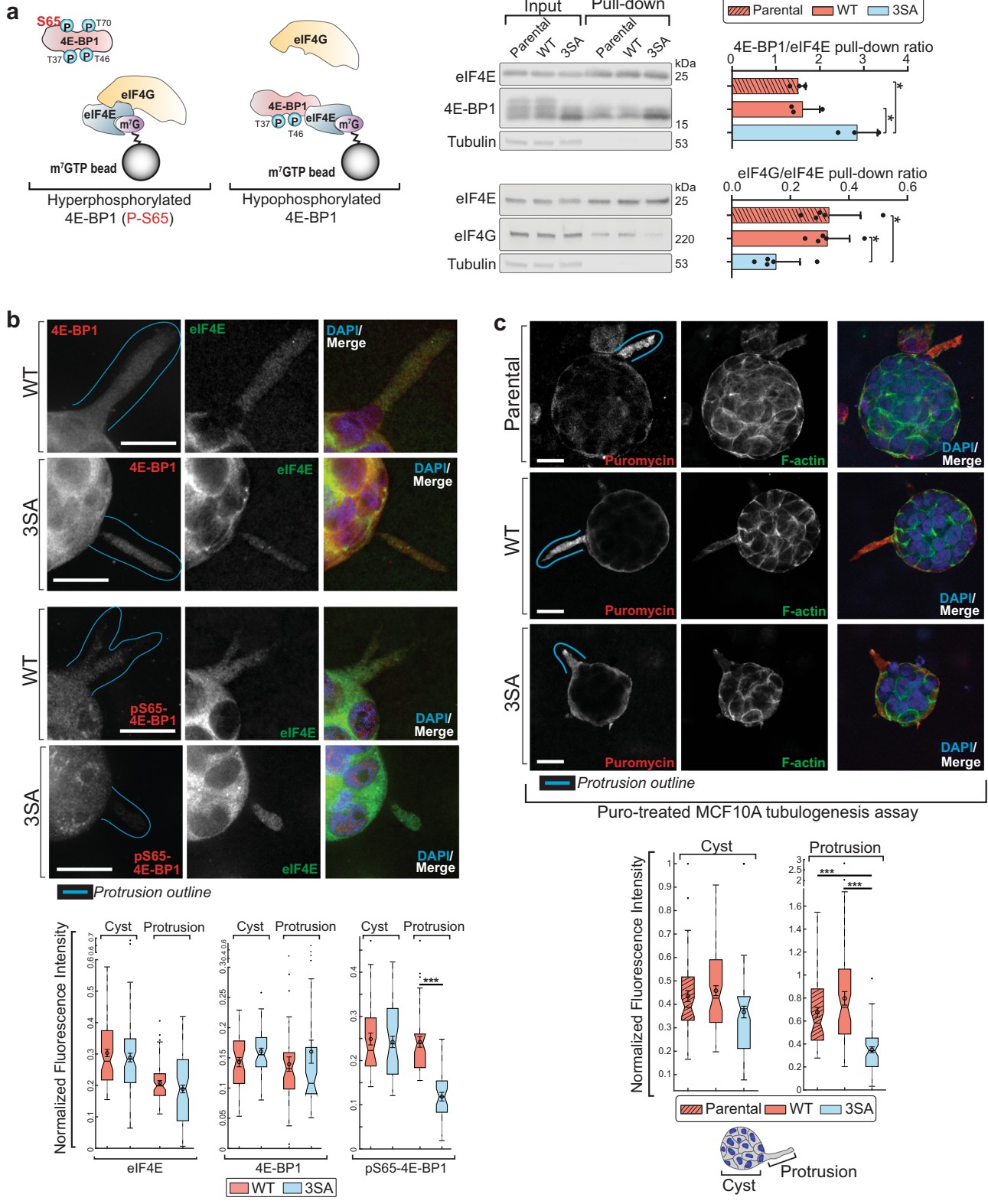

glands (Supplementary Data 1), supporting a general block in local translation. Unlike the 3D MCF10A tubulogenesis assay, 3D primary mouse organoid cultures do not depend on focal adhesions for migration. Elongation in these tubes is driven by intercalating migrating cells with compartmentalized Ras-PI3K activation and F-actin polymerization in leading protrusions[82]. Whether this depends on localized translation is not known, although we propose this may be the case, since we did observe

dysregulated 4E-BP1 phosphorylation and local translation in *Bad*[3SA] 3D organoids. Thus, localized translation potentially contributes to in vivo cell motility during mammary gland ductal elongation, and our study suggests that this is regulated by BAD.

In summary, these results identify BAD as a regulator of pubertal gland development, affecting localized mRNA translation and cell migration. BAD is normally phosphorylated in the pubertal mammary gland when ductal migration is extensive but

**Fig. 6 3D MCF10A cysts from 3SA have defective localized mRNA translation. a** Left: schematic for m[7]GTP pull-down assay to assess the interaction of eIF4E with 4E-BP1 or eIF4G. Middle: representative western blot from MCF10A 3D tubule lysates (day 7) from Parental, WT, 3SA showing levels of eIF4E, 4E-BP1, and eIF4G (Input). Lysates were subject to m[7]GTP pull-down and western blot shows levels of eIF4E, 4E-BP1, and eIF4G that were bound to m[7]GTP (Pull-down). Right: quantitation shows ratio of 4E-BP1 or eIF4G per eIF4E recovered from the m[7]GTP pull-down. 3SA (blue) shows increased 4E-BP1 and decreased eIF4G interactions with eIF4E. $n = 4$ independent experiments. **b** Top: representative immunofluorescence images of cyst protrusions from WT and 3SA in early tubulogenesis assays (day 3) stained with anti-eIF4E (green), either anti-4E-BP1 or anti-pS65-4E-BP1 (red) and DAPI (blue). The teal line is placed to outline subcellular protrusions subject to subsequent quantification analyses. Bottom: Tukey boxplots showing quantitation for normalized eIF4E, 4E-BP1, and pS65-4E-BP1 normalized fluorescence intensity in either the cyst or subcellular protrusion. Only pS65-4E-BP1 showed decreased staining, specifically in protrusions of 3SA. WT and 3SA show similar eIF4E and 4E-BP1 staining in the cyst and protrusions. WT $n = 35$, 3SA $n = 37$, from three independent experiments. **c** Top: representative immunofluorescence images of 3D MCF10A tubulogenesis assays treated with puromycin in culture, then processed for immunofluorescence and stained with anti-puromycin (red), phalloidin (F-actin, green) and DAPI (blue). Scale bars = 20 μm. Bottom: Tukey boxplots for normalized cyst and protrusion puromycin intensity show decreased puromycin staining in $Bad^{3SA}$ protrusions (blue). Parental $n = 57$, WT $n = 63$, 3SA $n = 60$ from three independent experiments. For Tukey boxplots, constriction indicates median, notch indicates 95% confidence interval, box edges are 25th and 75th percentiles, whiskers show extreme data points, "outliers" plotted as black dots, overlaid circle, and error bars are the mean ± SEM. For all $p$-values, ***$P < 0.001$, **$P < 0.01$, *$P < 0.05$. Statistical test details and exact $p$-values are provided in Supplementary Data 4. Source data are provided in the Source data file.

is not phosphorylated in the nulliparous adult. Whether BAD phosphorylation is aberrantly reactivated in breast carcinogenesis potentially facilitating metastasis, is unclear at this point. Altogether, this study provides a framework with which to query both normal and neoplastic mammary gland processes and link a non-canonical role of BAD and translation to mammary gland development and cancer pathophysiology.

## Methods

**Mouse line and breeding**. Animal procedures were performed in accordance with the guidelines and regulations set forth by the Canadian Council on Animal Care and approved by the University of Alberta Health Sciences 2 Animal Care and Use Committee (Protocol# AUP00000386). All mouse strains were in the C57BL/6J background and have been previously described[7,8,10].

**Whole-mounts imaging and morphological analysis**. Whole mount (WM) carmine staining of abdominal mammary gland #4 was done as described[94]. WMs slides were converted into digital images using an Epson Perfection scanner with a resolution of 2400 dpi, resulting in 10.5 μm pixel size RGB images. Image gamma was changed to 0.45 to contrast the epithelial tree against the stromal background. Morphological analysis was carried out on MATLAB (MathWorks). Terminal end buds (TEBs) on the leading edge and epithelial tree origin (nipple) were visually determined. To segment the epithelial tree, the RGB images were first converted to grayscale using principal component analysis transform. Contrast enhancement was then employed on the grayscale image using contrast-limited adaptive histogram equalization. Background image was then computed using a wide Gaussian (standard deviation 20 pixels) and subtracted from the contrast-enhanced image. The resultant image was binarized using Otsu's method in multithresh function. Additional morphological operations were carried out if necessary, to ensure the binary image matched the epithelial tree. From this epithelial tree binary image, boundary function (default shrink factor 0.5) was used to calculate epithelial tree area. Ductal extension was obtained from the longest distance between the nipple (tree origin) and tree boundary (in all whole mounts, ductal extension was towards the leading edge). The binary image was then skeletonized, and primary branches computed from the skeleton branchpoints. Manual refinement of the primary branches was performed to correct for artifacts such as detected 'branchpoints' due to independent ducts overlap. Dirichlet tessellation was computed using the branch points to determine the Voronoi polygon area occupied by each primary branch point within the epithelial tree area. For leading edge to lymph node (LN) distance measurements, LN, which was typically the darkest feature in the whole-mounts images, was segmented by looping though different threshold levels of multithresh function until a perfect LN boundary was obtained. The distance was computed from the resultant LN boundary centroid to the leading edge. The distance was considered positive if the epithelial leading edge was beyond LN centroid, otherwise, negative.

**Transplantation assay**. The transplant assay was performed as described[95]. Briefly, mammary gland epithelial fragments from 8-wk donor mice were implanted into cleared fat pads of mammary gland #4 of 3-wk recipient mice (Schematic Fig. 2d). Recipient glands were harvested 6 to 7-weeks post surgery and WMs carmine stained as aforementioned. WMs slides were imaged and donor epithelial tissue outgrowth assessed. Cleared, non-transplanted fat pads were used as controls for clearance efficiency of endogenous epithelium.

**Dissociation of mouse mammary epithelial cells and flow cytometry analysis**. 8-wk mouse mammary glands were minced using a sterile blade and subsequently digested in dissociation medium (20 mg Collagenase A and 10 mg Dispase II in 10 ml DMEM/F12) for 3 h at 37 °C with 120 rpm shaking. The dissociated tissue was spun down at $450 \times g$ for 10 min and then the supernatant was discarded. Preparation of a single cell suspension was done as detailed[45]. EasySep™ Mouse Epithelial Cell Enrichment Kit II (Stem Cell Technologies) was used to exclude non-epithelial (lineage-negative) cells following the manufacturer's instructions. For flow cytometry staining, cells were aliquoted for controls (unstained, single-stained, "fluorescence minus one" controls) and samples. Control and sample cells were accordingly incubated on ice for 10 min with PE-conjugated anti-CD24 (Stem Cell Technologies), FITC-conjugated anti-CD49f (Stem Cell Technologies) and Alexa Fluor® 647-conjugated anti-EpCAM (BD Biosciences). The dual combination of these surface markers (See Supplementary Fig. 6) were shown to have the least tendency of contaminating gated basal cells with non-epithelial cells, especially in the C57BL/6J background[96], which is the strain of mice used in our study. Flow cytometry was performed using the BD Accuri™ C6 and analyzed with FlowJo (version 10). Cell lineages were determined after pre-gating for single cells (Supplementary Fig. 6).

**Isolation of primary mouse mammary organoids**. Mammary organoids were prepared as described[32]. Briefly, mammary glands from 5-wk old mice were harvested and immediately minced with a scalpel (~40 times) under a laminar flow hood and digested for 30–40 min at 37 °C with 120 rpm shaking in 10 mL collagenase solution per mouse (Collagenase solution: 2 mg/mL collagenase (Sigma Cat# C2139), 2 mg/mL trypsin (Sigma Cat# T0303), 5% fetal bovine serum (FBS), 5 μg/mL insulin (Sigma Cat#I0516), and 50 μg/mL gentamicin (ThermoFisher Cat# 15710064) in DMEM/F12. Epithelial organoids were subsequently isolated through differential centrifugation and frozen in 10% DMSO, 20% FBS, and 70% DMEM/F12 freezing media for future use.

**Primary mouse organoid branching assay**. Mammary organoids were thawed, and freezing medium washed off with DMEM/F12 (2 times, 1200 rpm for 1 min). 50 μl of organoid suspension was aspirated onto a petri dish and the number of organoids counted under a tabletop microscope. From this tally, organoid density was adjusted to 4 organoids per μl. For 3D gel preparation, rat tail collagen-I (Corning Cat# 354236) was neutralized with 1 N NaOH and 10× DMEM (217:25:8 volume ratio, respectively), adjusted to 3 mg/mL final concentration (using DMEM/F12) and incubated for 1 h. Organoids were then embedded to a final density of 2 organoids per μl in pre-thawed 3 parts volume growth factor reduced Matrigel (Corning Cat# 354230) and 7 parts volume collagen-I solution, a ratio previously determined to be optimal for mouse branching organoids assay[31,32]. The organoid-gel suspension was gently plated in wells of a 12-well plate (50–100 μl per well), placed on top of a heating block (37 °C) to aid with gelation for a few minutes before placing the plate in a cell incubator (5% $CO_2$, 37 °C) for 45 min. Pre-warmed branching organoid medium (DMEM/F12 supplemented with 1× penicillin/streptomycin, 1× insulin-transferrin-selenium-X (GIBCO Cat# 51500), 10 ng/mL EGF (Peprotech Cat# AF100-15) and 2.5 nM FGF2(Sigma Cat# F0291)) was then added and replenished every 3 days.

**MCF10A tubulogenesis assay**. MCF10A cells were maintained in MCF10A growth medium[97] (DMEM/F12 supplemented with 5% horse serum, 20 ng/mL EGF (Peprotech Cat# AF100-15), 0.5 μg/mL Hydrocortisone (Sigma Cat# H-0888), 100 ng/mL Cholera toxin (Sigma Cat# C-8052), 10 μg/mL insulin (Sigma Cat# I-1882) and 1× penicillin/streptomycin). Stable ectopic expression of wild-type BAD (WT) or phosphomutant (3SA) was done in MCF10A $BAD^{-/-}$ ($BAD$ knockout)

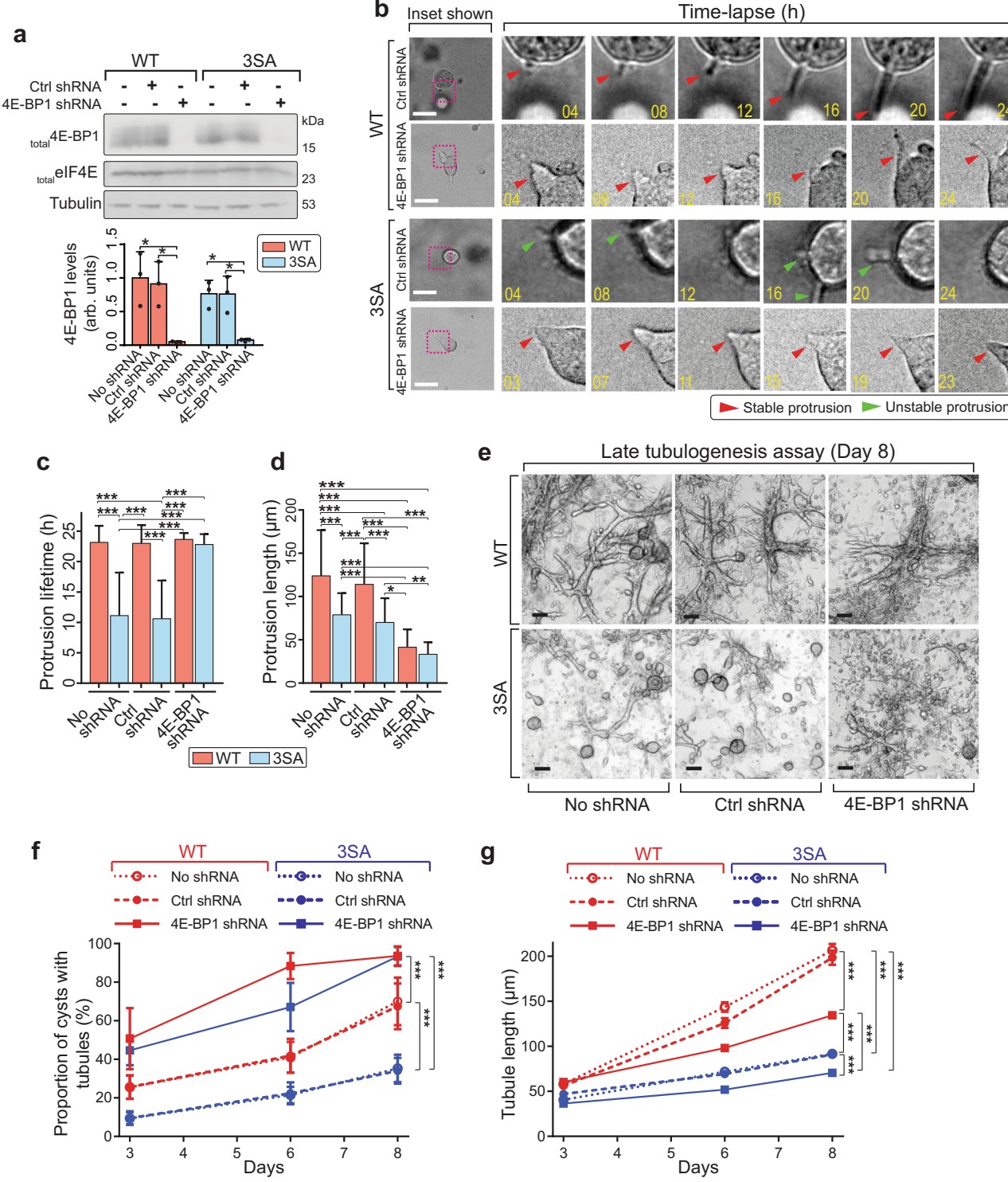

cells we previously described[13]. For 3D tubulogenesis assay gel preparation, bovine collagen-I (Advanced BioMatrix Cat# 5005) was neutralized and incubated for 1 h as detailed above for branching organoid assay. MCF10A single cells were then embedded (50 cell per µl) in pre-thawed 1 parts volume growth factor reduced Matrigel (Corning Cat# 354230) and 9 parts volume collagen-I solution (1 mg/mL final concentration). The cell-gel suspension was gently plated in wells of a 12-well plate (50–100 µl per well) placed on top of a heating block (37 °C) to aid with gelation for a few minutes before placing the plate in a cell incubator (5% $CO_2$, 37 °C) for 45 min. Pre-warmed MCF10A growth medium was then added and replenished every 3 days.

**m7GTP pull-down assay**. Whole gels containing MCF10A cysts and tubules were harvested on day 7 of the tubulogenesis assay in pull-down assay lysis buffer (20

mM Tris-HCl, 100 mM NaCl, 25 mM $MgCl_2$, 0.5% NP-40, freshly added protease, and phosphatase inhibitors (Roche Cat# 5056489001 and Cat# 4906837001, respectively)). Protein lysate was collected from the supernatant after $10,000 \times g$ 20 min centrifugation at 4 °C. To assess eIF4E:4E-BP1 association, for each sample, 20 µl of a 80% slurry of immobilized γ-aminophenyl-m7GTP beads (linked via a C10-spacer, Jena Bioscience Cat# AC-155) was washed three times in lysis buffer (500 × g for 30 s) and incubated overnight at 4 °C with the protein lysate with gentle rotation. Beads were then gently washed with the pull-down assay buffer (3 times, 500 × g for 30 s), bound proteins were eluted and analyzed on a western blot.

**SUnSET assay**. Mouse branching organoid and MCF10A tubulogenesis gels were incubated with 10 µg/mL puromycin (Sigma Cat# P8833) for 30 min. No puromycin and 15 min pre-treatment with 50 µg/mL cycloheximide (Sigma Cat#

**Fig. 7 3SA-mediated defects in protrusions and tubuologenesis are dependent on 4E-BP1. a** Top: representative western blots assessing 4E-BP1 levels in MCF10A WT and 3SA cell lines stably expressing 4E-BP1 shRNA. Lysates were generated from 3D tubulogenesis assays of indicated cell lines expressing either no shRNA (−), control non-specific shRNA (+ Ctrl shRNA), or 4E-BP1 shRNA (+ 4E-BP1 shRNA) and blots were probed for 4E-BP1 and eIF4E. Bottom: quantitation shows decreased 4E-BP1 protein levels in 4E-BP1 shRNA-expressing lysates (4E-BP1 shRNA). Data is mean ± SD from three independent experiments. **b** Representative bright-field time-lapse images for 3D tubulogenesis assays of WT (upper 2 panels) and 3SA (lower 2 panels) cells with either control knockdown (Ctrl shRNA) or 4E-BP1 knockdown (4E-BP1 shRNA). Imaging was started on day 3 of tubulogenesis assays. Yellow numbers in insets indicate timepoint (hrs) during imaging (for complete time-lapse images, see Supplementary Movie 3). Red arrowheads indicate stable protrusions and green arrowheads indicate unstable protrusions. Scale bars = 100 μm. WT Ctrl shRNA $n = 31$, WT 4E-BP1 shRNA $n = 26$, 3SA Ctrl shRNA $n = 31$, 3SA 4E-BP1 shRNA $n = 27$, from three independent experiments. **c** Quantitation of average protrusion lifetime shows that 3SA-destabilized protrusions are rescued by 4E-BP1 knockdown (blue + 4E-BP1 shRNA). WT no shRNA $n = 37$, WT Ctrl shRNA $n = 31$, WT 4E-BP1 shRNA $n = 26$, 3SA no shRNA $n = 37$, 3SA Ctrl shRNA $n = 31$, 3SA 4E-BP1 shRNA $n = 27$, from three independent experiments. Data are mean ± SD. **d** Quantitation of average protrusion length shows that 4E-BP1 knockdown decreases protrusion length in both WT and 3SA (orange and blue + 4E-BP1 shRNA). WT no shRNA $n = 37$, WT Ctrl shRNA $n = 31$, WT 4E-BP1 shRNA $n = 26$, 3SA no shRNA $n = 37$, 3SA Ctrl shRNA $n = 31$, 3SA 4E-BP1 shRNA $n = 27$, from three independent experiments. Data are mean ± SD. **e** Representative bright-field images for 3D tubulogenesis assays on day 7 of WT (upper) or 3SA (lower), expressing no shRNA, Ctrl shRNA or 4E-BP1 shRNA. Scale bars = 100 μm. Fields of view for WT $n = 16$, 3SA $n = 16$ from three independent experiments. **f** Percentage of cysts with branches over time for WT (red) or 3SA (blue). Line plots (mean ± SD) show 3SA-mediated low percentage of branching cysts is increased by 4E-BP1 knockdown (blue solid line). Percentage of branching cysts in WT is also increased by 4E-BP1 knockdown (red solid line). Fields of view analyzed used for all conditions, $n = 16$. **g** Quantitation of mean tubule length (mean ± SEM) for WT (red) or 3SA (blue). Line plots show 4E-BP1 knockdown diminishes mean tubule length in WT 3D tubules (red solid line). Three independent experiments, 2 replicates per experiment. For number of tubules, WT no shRNA, control shRNA, and 4E-BP1 shRNA $n = 132$, 124 and 146, respectively. 3SA no shRNA, control shRNA, and 4E-BP1 shRNA $n = 135$, 142, and 135, respectively. For all p-values, ***$P < 0.001$, **$P < 0.01$, *$P < 0.05$. Statistical test details and exact p-values are provided in Supplementary Data 4. Source data are provided in the Source data file.

C7698) were used as controls. Puromycinylated polypeptides were detected by western blot or immunofluorescence assays (see "Western blotting" and "Immunofluorescence" sections below).

**Cap-translation reporter gene assay.** Bicistronic fluorescent reporter gene plasmid pYIC (gift from Dr. Han Htun[98], Addgene plasmid Cat# 18673) with cap-dependent EYFP and IRES-dependent ECFP translation was used. MCF10A cells were electrotransfected using Amaxa nucleofector (program T020) and Lonza nucleofector Kit L. Cells were allowed to recover overnight and subsequently used in normal MCF10A tubulogenesis assays.

**4E-BP1 knockdown.** pLKO.1-puromycin vectors containing control shRNA (gift from Dr. David Sabatini[99], Addgene plasmid Cat# 1864) or 4E-BP1 shRNA (gift from Dr. Tommy Alain[100], Sigma Cat# TRCN0000040203) were packaged using MISSION® Lentiviral Packaging Mix kit (Sigma Aldrich Cat# SHP001) according to manufacturer's protocol. WT or 3SA MCF10A cells were infected with the lentivirus particles and selected using puromycin. Stable 4E-BP1 knockdown was confirmed using western blots (Fig. 7a).

**BAD immunoprecipitation.** Cells were treated as indicated and lysed in 1% CHAPS buffer (1% CHAPS, 150 mM NaCl, 50 mM Tris pH 7.4, 2 mM EDTA pH 8.0) supplemented with freshly added protease and phosphatase inhibitors. Lysates were incubated with anti-BAD antibodies (1.5 h) and recovered with Protein A Sepharose beads (Abcam, Cat# ab193256).

**Western blotting.** Mammary gland lysates were generated as described for RPPA lysates, below. Mouse branching organoids and tubulogenesis lysates were made by aspirating the gels 5 times in lysis buffer (27-gauge needle. Lysis buffer: 1% NP 40, 150 mM NaCl, 50 mM Tris-HCL pH 7.6, 5 mM EDTA, 1 mM EGTA, freshly added protease and phosphatase inhibitors) and soluble protein lysates were collected from the supernatant after $14{,}000 \times g$ 20 min centrifugation at 4 °C. Lysates were resolved on SDS-PAGE gels and transferred to PVDF membranes. Mouse or rabbit primary antibodies were used for western blots (Supplementary Data 4). HRP and IRDye-coupled secondary antibodies were used and scanned using Odyssey LI-COR Fc imager (LI-COR Biosciences) in the chemiluminescence, 700 nm and 800 nm channels. Band intensities were quantified with Image Studio™ version 5.2 (LI-COR Biosciences), which applies a local background subtraction determined for each band. All phosphorylation status was normalized to corresponding total protein. Uncropped blots are provided in the Source Data file.

**Immunofluorescence.** 100 μm thick cryosections were cut from OCT compound (ThermoFisher Cat# 23730572) embedded mammary glands or 3D gels which had been fixed (4% PFA) and stored in −80 °C[32]. The sections were washed with PBS (2×, 10 min), quenched with 50 mM NH$_4$Cl (30 min) and permeabilized with 0.5% Triton X-100 (30 min). For staining, slides were blocked (3% BSA in PBS, 1 h) and incubated with mouse or rabbit primary antibodies (Supplementary Data 4). Alexa-Fluor-488, Cy3 or Alexa-Fluor 647 tagged anti-rabbit and anti-mouse secondary antibodies were used. Depending on the experiment, phalloidin Alexa-Fluor-555 or

Alexa-Fluor-647 (Invitrogen) were used to stain actin. Nuclei were stained with DAPI.

**Imaging.** Mouse branching organoids and MCF10A tubulogenesis 3D gels time lapse imaging was done in brightfield (Zeiss AxioObserver.Z1 Microscope, ×20 air objective, 0.85 numerical aperture) on indicated days. All immunofluorescence imaging was acquired with Volocity software (PerkinElmer, USA) using a ×20 oil immersion objective on WaveFx spinning-disk microscope (Quorum Technologies, ON, Canada). The microscope was set up on an Olympus IX-81 inverted stand (Olympus, Japan), equipped with EM-CDD camera (Hamamatsu, Japan).

**Image intensity analysis.** Image intensity segmentation and analysis for the 3D gel immunofluorescence images was done as described[101]. Briefly, channel intensity was corrected for non-uniform background. Mean intensity or intensity ratios were then calculated within the defined protrusion or body mask. For SUnSET assay images, the body region was restricted to 10 μm around the body mask (Supplementary Fig. 9c) since the body central region was mostly devoid of puromycin intensity in all assays and genotypes. To quantify localized translation, we computed the ratio of puromycin intensity in the protrusion region over a size-matched region within the cyst body (Supplementary Fig. 9c).

**Mass spectrometry and RPPA lysates screening.** Using five independent mice for each screen and each group, mammary gland lysates from 5-wk $Bad^{+/+}$ and $Bad^{3SA}$ animals along with 4-wk $Bad^{+/+}$ were generated (3 groups, 2 screens, total mice = 30). The #4 mammary glands were harvested and immediately snap frozen in liquid nitrogen after lymph node removal. Samples were then lysed with a hand homogenizer (VWR Cat# 47747-370). The lysis buffer for mass spectrometry (MS) samples was: 1% NP 40, 150 mM NaCl, 50 mM Tris-HCL pH 7.6, freshly added protease and phosphatase inhibitors (Roche Cat# 5056489001 and Cat# 4906837001, respectively). The lysis buffer for RPPA samples was: 1% Triton X-100, 50 mM HEPES, pH 7.4, 150 mM NaCl, 1.5 mM MgCl$_2$, 1 mM EGTA, 100 mM NaF, 10 mM Na pyrophosphate, 1 mM Na$_3$VO$_4$, freshly added protease and phosphatase inhibitors (Roche Cat# 5056489001 and Cat# 4906837001, respectively). Soluble protein lysate was collected from the supernatant after $16{,}000 \times g$ 20 min centrifugation at 4 °C. RPPA sample lysates were then diluted to 1 μg/μl (using 4X sample buffer: 40% glycerol, 8% SDS, 0.25 M Tris-HCL, pH 6.8 and 10% β-mercaptoethanol) and submitted to the MD Anderson RPPA Core Facility (University of Texas) for screening and initial analysis.

For mass spectrometry analysis, homogenates (20 μg of protein per sample) were run on 10% sterile-filtered SDS polyacrylamide gels, and subject to in-gel trypsin digestion[102]. Whole gel-lanes were excised using a scalpel and sectioned into 10–11 bands, each of which was further sectioned into 1 mm cubes. Cubes corresponding to an individual band were placed in a 96-well plate and subject to trypsinization. Extracted peptides from each well were concentrated and resuspended in 60 μL of 0.2% formic acid in HPLC-grade water. Peptides were analyzed by LC-MSMS using a Thermo Easy nLC-1000 in tandem with a Q-Exactive benchtop orbitrap mass spectrometer. 5 μL of sample from each well was subject to a 75-min gradient (0–45% buffer B; buffer B = 0.2% formic acid in acetonitrile) on a 2 cm Acclaim 100 PepMap Nanoviper C18 trapping column with

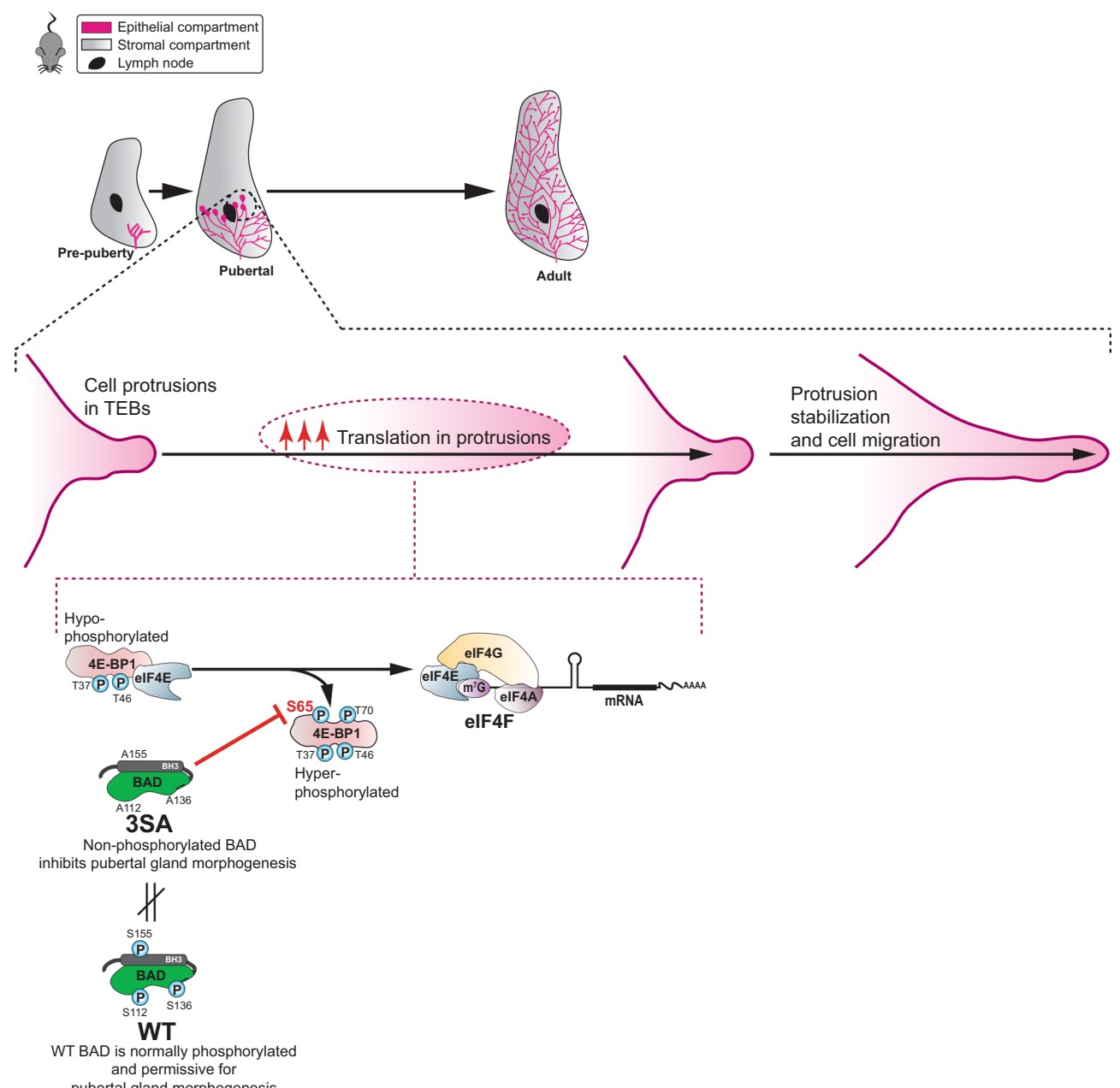

**Fig. 8 Schematic model for the role of BAD in normal pubertal mammary development.** BAD is normally phosphorylated in the pubertal mammary gland and this is permissive for 4E-BP1 hyperphosphorylation and subsequent upregulation of localized translation for efficient cell migration and epithelial tree elongation. Preventing BAD phosphorylation (3SA) inhibits hyperphosphorylation of 4E-BP1 that impairs cell migration and tree elongation.

a New Objective PicoChip reverse-phase analytical LC column, with spray voltage set to 2700 V.

**Mass spectrometry data processing**. Raw data files comprising an entire gel lane were grouped and searched using Proteome Discoverer (PD) 1.4.1.14's SEQUEST search engine using a reviewed, non-redundant complete *Mus musculus* proteome retrieved from UniprotKB on October 16, 2015[102]. Magellan storage files from all $n = 15$ samples were opened in unison using the PD 1.4.1.14 software package, with data filtered to show only proteins identified with at least 2 medium confidence peptides (FDR < 0.05 per peptide). Relative protein abundance was determined using proteins' extracted ion chromatograms (EICs) using PD 1.4.1.14's 'Precursor Ion Areas' module, in relation to the summation of all EICs from a given sample. Data were exported to Microsoft Excel.

**Hierarchical clustering and random forest classification**. Both MS and RPPA protein profile unsupervised hierarchical clustering was performed on MATLAB (clustergram function). Samples distance metric was set to correlation, and average distance between all pairs of objects in any two clusters was used to create

dendrograms agglomerative hierarchical cluster tree. All total/phosphoproteins in RPPA were analyzed. For MS protein profiles, proteins with zero ion intensity were assigned half of the global minimum ion intensity in the dataset. Then, for each group, proteins that initially had zero ion intensity in more than two samples were filtered out (799 proteins left).

Random forest classification was computed on MATLAB (Ensemble of decision trees based TreeBagger function). Briefly, 4-wk $Bad^{+/+}$ and 5-wk $Bad^{+/+}$ protein profiles were used as training data (using 5000 decision trees) and all samples were subsequently scored (probability of being classified as 4-weeks or 5-weeks $Bad^{+/+}$ protein profile) based on the trained model.

**Gene ontology and pathway analysis**. For RPPA data, differential candidates were derived from analysis of variance $p$-values ($p < 0.05$), indicative of significantly different total/phosphoproteins levels amongst the groups. These candidates (Supplementary Data 2) were subjected to Pathway analysis scoring using the Enrichr web application[103].

For mass spectrometry data, differential protein expression between experimental conditions was determined by performing pair-wise analyses of

relative protein abundance observed between experimental conditions. Comparisons were limited to proteins observed in at least $n = 3$ of 10 samples. Two-tailed heteroscedastic $T$-tests were performed on relative protein abundances between experimental conditions, with resultant $p$-values being uploaded to qvalue. princeton.edu to generate false-discovery rates. Additionally, differences in proteins' relative abundance between experimental conditions was determined by generating log2 ratios of experimental conditions' proteins' average relative abundance, with proteins uniquely observed given an arbitrary log2 value of ±10. Significance for differential protein abundance between experimental conditions was determined using either $p < 0.05$ and/or a log2 ratio of ±2 (indicating a 4-fold change). Proteins determined to be preferentially abundant for a given experimental condition were grouped and subject to GO analysis using the Enrichr web application.

**Graphing, statistics, and reproducibility**. Graphing was done on GraphPad prism (Graphpad Software, La Jolla, CA, USA) and MATLAB. All error bars in bar graphs are represented as mean ± standard deviation. For all Tukey boxplots, notch indicates 95% confidence interval around the median line, box edges are 25th and 75th percentiles, whiskers extend to extreme data points excluding "outliers" (individual "outliers" are plotted as black dots). Overlaid circle and error bars are the mean ± standard error of mean. For WMs morphological analysis bar plots, all western blots and RPPA bar plots: two-tailed unpaired $t$-test was used for 2 groups, while ANOVA and subsequent Tukey-Kramer's post hoc test was used for multi-comparison of 3 or more groups. For all other imaging data: two-sided Wilcoxon rank-sum pair-wise test was used for 2 groups, while Kruskal–Wallis test followed by Dunn's pair-wise post-hoc test was used for multi-comparison of 3 or more groups. Significant $p$-values are shown in figures. All statistical tests and subsequent exact $p$-values are shown in Supplementary Data 4.

**Reporting summary**. Further information on research design is available in the Nature Research Reporting Summary linked to this article.

## Data availability

The mass spectrometry proteomics data have been deposited in the ProteomeXchange Consortium via the PRIDE partner repository[104] under the dataset identifier PXD014347. The comparative proteomic data analysis has been included in Supplementary Data 1 (*Mus musculus* proteome retrieved from UniprotKB 'release-2015_10' version on October 16, 2015). RPPA raw data is included in Supplementary Data 2. All data used in this study are available upon reasonable request. Source data are provided with this paper.

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

## Acknowledgements

We are grateful to all Goping lab members for their valuable discussions and contribution. We thank Dr. Tommy Alain for graciously sharing 4E-BP1 shRNA plasmid and Drs. Elena Posse de Chaves, Thomas Simmen, and Hanne Ostergaard for their generous gift of antibodies. ISG acknowledges the Lilian McCullough Chair in Breast Cancer Research. This work was supported by operating grants from the Alberta Cancer Foundation and Canadian Institutes of Health Research to ISG.

## Author contributions

I.S.G. and D.A.U. conceived the study. J.M.G., N.T., and I.S.G., designed research. J.M.G., N.T., R.K., N.P., V.P., R.M., L.F.Z., and D.A.K. performed the experiments. N.D. provided mouse strains. N.D. and N.S. provided expertise. J.M.G. and D.A.K. analyzed the data. I.S.G. and R.P.F. directed research. J.M.G. and I.S.G. wrote the paper.

## Competing interests

The authors declare no competing interests.
