## [Peer Review File · Nature Communications]

Reviewers' Comments:

Reviewer #1:

Remarks to the Author:

The manuscript entitled "BAD regulated mammary gland morphogenesis by EP-BP1-mediated control of localized translation" describes abnormal pubertal mammary gland development in mice that carrying a specific point mutation that prevents BAD phosphorylation. Moreover, the authors utilized proteomic approaches to identify 4E-BP1 and the main target misregulated in the absence of BAD phosphorylation. Using organoid cultures, 4E-BP1 gene rescue stabilized organoid protrusions.

This is a very interesting and well writing manuscript. The figures are well structured and mostly stand alone.

One of the limitations of the study is the lack of mammary epithelial cell analysis, to define which cell type was the most affected in the absence of BAD phosphorylation. In addition, a more in-depth investigation/discussion about why the mammary developmental defect in only present during the early stages of puberty, and why such developmental defect go away after puberty, during pregnancy and during involution would highlight the importance of such mechanisms on overall mammary development. Collectively, these limitations take away some of the enthusiasm about the manuscript.

Point to be addressed:

1 – Please indicate pvalues and number of replicates in all figures

2- replace "significantly increased", or "decreased in comparison" with more quantitative measurement of differences (percentage, fold change, etc)

3- Immune infiltration, ECM remodeling , epithelial differentiation block are all aspects that can influence duct elongation and branching morphogenesis. Such pathways can also be altered by abnormal epithelial cells in transplantation assays. Thus, investigation of alterations to these pathways in BAD3SA mammary glands are required to fully understand the role of BAD, and 4E-BP1, on mammary morphogenesis.

3 – On Fig.2, BAD3SA MECs were utilized for mammary fatpad transplants. Were these MECs harvested from a mouse with 5 weeks of age or older? Would transplantation of BAD3SA MECs harvested from an older animal, or a female after pregnancy, develop normally in BAD WT fatpad?

4 – On Fig.3, the authors demonstrated that BAD3SA organoids phenocopied the defective ductal elongation. It would be interesting to investigate whether the addition of pregnancy hormones to BAD3SA organoids corrects the duct elongation phenotype, like shown during pregnancy in vivo

5 – the authors speculate that BAD could play a role in tumorigenesis. Is BAD phosphorylation (or lack of it) and mutations frequent in breast oncogenesis?

Reviewer #2:

Remarks to the Author:

Overall comments:

The article describes the morphogenesis related mechanism in mammary gland development using skilled techniques, but it is too superficial to describe the role of BAD in puberty gland development without considering their defined role in apoptotic signaling. The experimental design is not clear, as there is no overexpression BAD in mouse or human cell line included as a control.

Besides, it is contradictory that the authors examined BAD phosphorylation at different residues using different techniques and came up with a single conclusion. One immortalized human cell line included in the study is not sufficient to reflect and prove the phenomenon in the puberty stage. Besides, this article only imparted BAD as a regulatory molecule during puberty gland development but didn't provide more details about different stages in mammary gland development. Additionally, the authors proposed that "It is possible that BAD phosphoregulatory mechanisms in puberty are aberrantly reactivated in breast carcinogenesis potentially facilitating metastasis", however, the assumptions are not supported by the data presented. Also, several caveats of the presented data, a few described in the specific comments below, attenuate the significance of their findings and the quality of the manuscript. The current state of the manuscript required a substantial amount of revision and proofread to meet publication standards.

Major comments;

- What is the rationale to investigate BAD protein's role in mammary gland morphogenesis? How authors hypothesised that BAD3SA may play a role in mammary gland morphogenesis or development?? not clear from the introduction section.
- Previously reported that unphosphorylated BAD sequesters BCL-2, which results in BAK/BAX activation and apoptosis. Therefore, why here BAD3SA not examined for cell survival-related functions? What happens to BAD3SA protein interaction with BCL-2 protein?
- Experimental evidence is not adequate for the mechanistic association between BAD and 4EBP-1 regulation. Seems overstated. More specific data needed.
- Current findings of the manuscript suggest BAD protein may serve as a facilitator for the mammary gland development, not a regulator.

General comments;

1. Page 4 line 68-77

Why did the author use phosphor-BAD antibodies at different residuals for WB and IF? pBAD (ser112) and pBAD (Ser136) were phosphorylated via different signaling pathways, so the conclusion that p-BAD modulate the pubertal process is hasty.

2. Page 5 line 81-84

Why did the author choose BADS155A instead of BADS136A or BADS112A knock-in genetic mouse since the author showed significant changes in pBAD (ser112) and pBAD (Ser136) for WB and IF respectively in Figure 1? How different BAD phosphorylation including BADS155A, BADS136A and BADS112A influence on ductal elongation is unknown.

3. Page 6 line 113-114

In figure 3F, to prove the ectopically expressed human wild-type (WT) BAD or BAD3SA, it is necessary to detect Ser112,136 and 155 BAD phosphorylation protein.

4. Page 7 line 117-118

Is there STR and mycoplasma free identification for MCF10A? I doubt there is something wrong with this cell line (contamination?).

5. In general, blot quality is not adequate. Densitometry analysis is needed.

6. Page 9 line 159-163

PI3K-AKT-mTOR signaling is mainly affecting the phosphorylation of BAD at Ser136, but why didn't the author perform Western Blot on pBADSer136 instead of pBADSer112.

7. Page 13 line 248-249

mTOR can phosphorylate BAD at Ser136 by downstream substrate p70-S6K, it is necessary to study its expression to exclude the upstream signaling interference.

Reviewer #3:

Remarks to the Author:

The manuscript by Goping and colleagues describes the exciting finding that Bcl-2 family member, BAD, may play a role in mammary gland development through the regulation of localized

translation by controlling the phosphorylation of the inhibitor of cap-dependent translation, 4E-BP1. Little is known about the connection of these proteins so the finding could open new doors in the study of 4E-BP1 biology and regulation of cap-dependent translation. Thus, based on its novelty and likely impact on the field, the manuscript is a good fit for publication in Nature Communications; however, I will raise a few issues and points for consideration that should be addressed prior to publication.

1. In Figure 1A, the authors provide expression data for pS112-BAD and total BAD via Western blot and its quantitation. Based on visual inspection of the blot, the quantitation does not appear to be correct. For example, the bands for total BAD in the puberty and pregnant samples look much more similar than the graphing would imply. Can the authors comment a bit more on how the quantitation was performed?
2. In Figure 6A, the authors performed a cap pulldown assay to monitor eIF4E-4EBP1 binding. Because cap-dependent translation is initiated upon 4E-BP1 release and eIF4G binding to eIF4E through competition for the same binding site, the authors should also show data for eIF4G which should exhibit an opposite trend as 4E-BP1.
3. While the SUNSET assay demonstrates a measure of total translation, the authors imply that the effects are due specifically to regulation of cap-dependent translation. Assays should be performed to demonstrate this, such as the well-known luciferase-based cap-dependent/independent translation assay. It would also be nice to see data demonstrating specific effects on eIF4E protein-protein interactions in the protrusions using the PLA assay as has been published for eIF4E interactions with 4E-BP1 and eIF4G.
4. As the authors indicated, additional kinases aside from mTORC1 have been demonstrated to phosphorylate 4E-BP1, namely CDK1 and CDK4. Have the authors explored any connections between these kinases and their findings regarding BAD? Perhaps some discussion of this could be included.

Reviewer #1 (Remarks to the Author):

1. *This is a very interesting and well writing manuscript. The figures are well structured and mostly stand alone.*

Author response: We thank the Reviewer.

2. *One of the limitations of the study is the lack of mammary epithelial cell analysis, to define which cell type was the most affected in the absence of BAD phosphorylation..... Immune infiltration, ECM remodeling , epithelial differentiation block are all aspects that can influence duct elongation and branching morphogenesis. Such pathways can also be altered by abnormal epithelial cells in transplantation assays. Thus, investigation of alterations to these pathways in BAD3SA mammary glands are required to fully understand the role of BAD, and 4E-BP1, on mammary morphogenesis.....On Fig.2, BAD3SA MECs were utilized for mammary fatpad transplants. Were these MECs harvested from a mouse with 5 weeks of age or older? Would transplantation of BAD3SA MECs harvested from an older animal, or a female after pregnancy, develop normally in BAD WT fatpad?*

Author response: The reviewer raises the point that one must identify which cell type was most affected in the absence of BAD phosphorylation. We couldn't agree more and indeed we thoroughly investigated this and present definitive results. Key experiments are both our epithelial tissue transplant assay and epithelial cell-autonomous 3D organoid assay. Mammary

gland transplantation is still considered the gold standard for delineating whether epithelial cells or stromal components contribute to phenotypes of whole-body genetically manipulated animals. Our transplant assay (Figure 2d-h) shows that our phenotype follows the epithelium since epithelial tissue from *Bad*^{+/+} animals grows normally in *Bad*^{3SA/3SA} stroma. Furthermore, we recapitulate the phenotype in both mouse and human epithelial 3D organoids. This robust system confirms the effect is truly epithelial cell autonomous since the elongation occurs within defined ECM components (Matrigel/collagen) without the need for stromal cells (eg. immune cells as the reviewer asks). In response to the query of the age of the mice for the MEC harvest for the transplants, the donor tissue was harvest from mice that were 8wks of age. See manuscript line 411 “The transplant assay was performed as described⁹². Briefly, mammary gland epithelial fragments from 8-wk donor mice were implanted into cleared fat pads of mammary gland #4 of 3-wk recipient mice (Schematic Fig. 2d)”. We trust this answers in part the Reviewer’s question of “*would transplantation of BAD3SA MECs from an older animal develop normally in BAD WT fatpad?*” Our data indicates that the defect is not specific to the tissue age at harvest, but is driven by the genotype of the MECs, which affects the process of ductal morphogenesis.

3. *In addition, a more in-depth investigation/discussion about why the mammary developmental defect in only present during the early stages of puberty, and why such developmental defect go away after puberty, during pregnancy and during involution would highlight the importance of such mechanisms on overall mammary development. Collectively, these limitations take away some of the enthusiasm about the manuscript.*

Author response: This is a good suggestion and we agree. We have expanded our discussion of the significance of the pubertal delay in the context of other mouse genetic mutations that phenocopy our data. Starting at line 303: “Finally, the *Bad*^{3SA} effect of defective pubertal ductal elongation is transient, as by adulthood, mice possess fully functional mammary glands. As mentioned previously, similar pubertal delay phenotypes have been reported in *in vivo* depletion studies of mTORC1, which is consistent with our model where *Bad*^{3SA} disrupts a key mTORC1 target, 4E-BP1. Ductal pubertal delay has also been reported in FGFR-null mammary glands⁶⁹, suggesting that BAD phosphorylation is downstream of FGFR signalling. This is consistent with our study, as FGF induces tubulogenesis in our experimental mouse 3D organoid cultures, which is blocked by *Bad*^{3SA}. Furthermore, since FGFR signaling is responsive to estrogen and progesterone⁷⁰, this provides an explanation for the pubertal-specific effect of *Bad*^{3SA}. Additionally, stromal depletion of Sharpin⁷¹ also phenocopies *Bad*^{3SA}. In this case, loss of Sharpin decreases ECM collagen stiffness, diminishing integrin signalling. We speculate that by reducing the translation of focal adhesion components such as paxillin and actin, *BAD*^{3SA} would similarly disable integrin function”

4. *Please indicate pvalues and number of replicates in all figures.*

Author response: This has been done and we have added Sup. Table 4, which includes all p values from all figures

5. *replace “significantly increased”, or “decreased in comparison” with more quantitative measurement of differences (percentage, fold change, etc)*

Author response: We have done this.

6. On Fig. 3, the authors demonstrated that *BAD3SA* organoids phenocopied the defective ductal elongation. It would be interesting to investigate whether the addition of pregnancy hormones to *BAD3SA* organoids corrects the duct elongation phenotype, like shown during pregnancy *in vivo*

Author response: This is an interesting suggestion, and we performed the experiments. The addition of pregnancy hormones, however, did not rescue the ductal elongation phenotype. As such, it was not included in the study due to length constraints, however, we are happy to share this data with the Reviewer (see below):

MCF10A tubulogenesis assays show pregnancy hormones do not rescue 3SA defects. Left) Representative bright-field images for 3D MCF10A tubulogenesis assays, from day 8, of WT (top) and 3SA (lower). Cells were control-treated, or treated with prolactin (10nM) or somatotropin (10nM) throughout the experiment period. Right) Quantitation of mean tubule length from 3 independent experiments. For Tukey boxplots, constriction indicates median, notch indicates 95% confidence interval, box edges are 25th and 75th percentiles, whiskers show extreme data points, 'outliers' plotted as black dots, overlaid circle and error bars are the mean \pm SEM.

7. The authors speculate that *BAD* could play a role in tumorigenesis. Is *BAD* phosphorylation (or lack of it) and mutations frequent in breast oncogenesis?

Author response: This is a very interesting question. In fact, *BAD* phosphorylation has been shown to be significantly associated with aggressive disease and significantly associated with the most aggressive and motile breast cancer subtype (Triple Negative Breast Cancer—TNBC) (Boac, Bernadette M et al. "Expression of the *BAD* pathway is a marker of triple-negative status and poor outcome." Scientific reports vol. 9, 17496. 25 Nov. 2019, doi:10.1038/s41598-019-53695-0). This correlative clinical information is very intriguing. Importantly, this highlights the importance of our functional studies that demonstrate a causal relationship between non-P-*BAD* and inhibited cell motility, *in vivo*. Importantly, this sets the stage for our future studies where we will be exploring molecular links between P-*BAD* and pathological cancer metastasis.

With respect to mutations, *BAD* is not highly mutated in breast oncogenesis. Across 74,247 tumor samples from multiple cancer types in cBioPortal, *BAD* was mutated in 0.57% (31.3% for TP53, as a reference). In 9131 breast cancer samples, 0.67% (32.2% for TP53) had

alterations but only 2 patients (0.02%) were characterized by missense mutations. Moreover, BAD is not annotated as a cancer census gene in COSMIC. Thus, BAD phosphorylation is the more clinically relevant modification to explore in studies of breast oncogenesis, which again emphasizes the relevance of our study examining BAD-phosphorylation effects *in vivo*.

Reviewer #2 (Remarks to the Author):

Overall comments:

1. *The article describes the morphogenesis related mechanism in mammary gland development using skilled techniques, but it is too superficial to describe the role of BAD in puberty gland development without considering their defined role in apoptotic signaling.... Previously reported that unphosphorylated BAD sequesters BCL-2, which results in BAK/BAX activation and apoptosis. Therefore, why here BAD3SA not examined for cell survival-related functions? What happens to BAD3SA protein interaction with BCL-2 protein?*

Author response: We originally conducted 2 unbiased proteomic screens that found no significant difference in apoptosis signatures via mass spectrometry GO analysis (Sup. Fig. 4D) and weak ranked significance of RPPA pathway analysis (Sup. Fig. 5C, ranked 12 of 20). Instead, the most significant hits identified Focal Adhesion, mTOR pathway and actin-binding molecular functions and therefore, these cellular programs were prioritized. Reviewer 2 makes the reasonable suggestion that apoptosis should be specifically examined. We had indeed done this, and now include the data (Sup Fig. 5; Sup. Movie 2). The new text starts at line 155: “BAD^{3SA} has been shown to induce apoptosis⁷, so we examined whether apoptotic signalling mediated protrusion destabilization. As expected, cleaved caspase 3 was detected in the TEBs of the pubertal mammary gland²⁰ but was not significantly increased in *Bad*^{3SA} (Sup. Fig. 5a). Additionally, blocking caspase activity with the pan caspase inhibitor zVAD-fmk did not rescue protrusion defects in BAD^{3SA} tubulogenesis assays (Sup. Fig. 5b,c; Sup. Movie 2). BAD stimulates apoptosis by binding to anti-apoptotic Bcl-XL⁴². Although BAD^{3SA} bound strongly to Bcl-XL in the 3D culture system, disrupting this interaction with the BH3-mimetic ABT-737 (Sup. Fig. 5d) did not alter the ability of BAD^{3SA} to inhibit protrusion stability (Sup. Fig. 5b). Thus, the mechanism whereby BAD^{3SA} inhibited ductal elongation did not require Bcl-XL interaction or caspase activity and was independent of apoptosis.”

2. *The experimental design is not clear, as there is no overexpression BAD in mouse or human cell line included as a control.*

Author response: We do not agree that BAD-expressing controls are missing. Firstly, overexpression of BAD in the mouse lines as the Reviewer suggests, is not the proper control. As stated in the text, starting at line 65: “We used 3 genetic engineered mouse models to explore the role of BAD in postnatal mammary gland development; knock-out *Bad*^{-/-}, knock-in *Bad*^{S155A} and *Bad*^{3SA} where 3SA indicates alanine substitutions at S112/136/155 of the endogenous *Bad* allele (gene in italics and protein in all uppercase)^{7, 8, 10}”. The proper control for these strains is the wild-type animal that has the wild-type endogenous *Bad* allele (*Bad*^{+/+}). This is the control we used and is clearly indicated in the text, figures and figure legends as *Bad*^{+/+}. Secondly, for the human cell line data, we included 2 positive BAD-expressing controls and this is indicated in the text (starting line 114) “We knocked-out expression of endogenous BAD (human gene in all cap italics) and ectopically expressed human wild-type (WT) BAD or BAD3SA (herein, referred to

as WT and 3SA, respectively). All experiments were conducted alongside an additional control of parental MCF10A cells". All controls are clearly stated in the text and labeled in the figures and legends.

3. Besides, it is contradictory that the authors examined BAD phosphorylation at different residues using different techniques and came up with a single conclusion..... 1. Page 4 line 68-77 Why did the author use phosphor-BAD antibodies at different residuals for WB and IF? pBAD (ser112) and pBAD (Ser136) were phosphorylated via different signaling pathways, so the conclusion that p-BAD modulate the pubertal process is hasty.

Author response: Indeed, we were able to examine BAD phosphorylation at different residues, because phosphorylation of all 3 serine sites is temporally coordinated (ref 26 in manuscript), where S136 phosphorylation primes phosphorylation of S112 and S155, and S155-dephosphorylation enhances dephosphorylation of S112 and S136. Since our study was examining the effects downstream of BAD phosphorylation, positivity with either P-S112, P-S136 or P-S155 would indicate phosphorylation of all 3 residues. The commercial antibodies that we used were optimized by us for each application of western blotting or microscopy/immunofluorescence as we had an ideal negative control of *Bad*^{3SA/3SA} knock-in tissue (see Figs 1a, 3f, 5d-e for anti-P-S112 blots and Fig 3f and Sup Fig3a for anti-P-S112 and anti-P-S136 immunofluorescence). Therefore, we routinely used anti-P-S112 for western blotting and immunofluorescence and anti-P-S136 for immunofluorescence. In response to the Reviewer's request, we have included western blots of the same lysates using both anti-BAD_P-S112 and anti-BAD_P-S136 (see Fig. 3f). Thus, given that phosphorylation of those serine residues are coordinately linked and that we were examining effects downstream of BAD phosphorylation, we were justified in postulating, as the Reviewer says, a "single conclusion".

4. One immortalized human cell line included in the study is not sufficient to reflect and prove the phenomenon in the puberty stage.

Author response: If the Reviewer is suggesting that the "phenomenon in the puberty stage" is not "proven", we respectfully disagree. Firstly, the phenotype is described *in vivo* with whole animals and presented in two related but independent genotypes of *Bad*^{3SA/3SA} and *Bad*^{S155A/S155A}, providing convincing evidence of physiological pubertal delay at both the morphological level (Fig. 2a-c) and the proteomic level (Sup. Fig. 4). Two independent organoid models that mimic pubertal gland morphogenesis were used to validate the BAD-dependent phenotype; (i) *ex vivo* mouse organoids (Fig. 3a-d) and (ii) human MCF10A 3D tubulogenesis assays (Fig. 3e-i). Together, this strongly validates the phenotype. Secondly, the Reviewer stated that only "one immortalized human cell line included in the study is not sufficient". To date, the MCF10A cell line is the only immortalized non-cancerous human breast epithelial cells which have been extensively used to study tubulogenesis in 3D models (refs Guo et al. 2012, PNAS 109: 5576; Krause et al. 2012, Tissue Eng. 18, 520; Barnes et al. 2014, PloS one 9, e93325; Dhimolea et al., 2010, Biomaterials 31, 3622; Accornero et al. 2012, PloS one 7: e44982). There have been efforts to utilize human induced pluripotent stem cells to model tubulogenesis, although this approach is still in its infancy and beyond the scope of our study (ref Qu, et al., 2017, Stem cell reports 8: 205. doi:10.1016/j.stemcr.2016.12.023).

5. Besides, this article only imparted BAD as a regulatory molecule during puberty gland development but didn't provide more details about different stages in mammary gland development.

Author response: We would like to draw the Reviewer's attention to Sup. Fig. 2, where we in fact do show different developmental stages in the adult, pregnant and involuting mammary glands.

6. Additionally, the authors proposed that "It is possible that BAD phosphoregulatory mechanisms in puberty are aberrantly reactivated in breast carcinogenesis potentially facilitating metastasis", however, the assumptions are not supported by the data presented.

Author response: Indeed, this is simply a speculative statement in the Discussion exploring possible relevance in pathophysiology. We have modified the statement to, line 357: "BAD is normally phosphorylated in the pubertal mammary gland when ductal migration is extensive but is not phosphorylated in the nulliparous adult. Whether BAD phosphorylation is aberrantly reactivated in breast carcinogenesis potentially facilitating metastasis, is unclear at this point."

7. Also, several caveats of the presented data, a few described in the specific comments below, attenuate the significance of their findings and the quality of the manuscript. The current state of the manuscript required a substantial amount of revision and proofread to meet publication standards.

Author response: Any editorial suggestions for revisions and proofreading will be addressed.

8. What is the rationale to investigate BAD protein's role in mammary gland morphogenesis? How authors hypothesised that BAD3SA may play a role in mammary gland morphogenesis or development?? not clear from the introduction section.

Author response: BAD is a prognostic indicator for breast cancer patient survival, although how, or even whether BAD regulates mammary gland homeostasis is unknown. Thus, we decided to investigate mammary gland development in order to gain a physiologically relevant understanding of the role of BAD in breast cell biology. This rationale is stated in the introduction as follows (starting line 29): "In the breast, BAD is a prognostic marker for survival of breast cancer patients (12), and modulates mitochondrial metabolism and sensitivity to taxane chemotherapy in vitro (13, 14). Intriguingly, BAD is differentially expressed during mammary gland development in the mouse and deciphering this may shed light on pathophysiology as aberrant reactivation of these developmental pathways defines breast carcinogenesis (16-18)."

10. Experimental evidence is not adequate for the mechanistic association between BAD and 4EBP-1 regulation. Seems overstated. More specific data needed.

Author response: The causal link between BAD and 4E-BP-1 regulation are clearly supported by 3 independent assays. These results all demonstrate that BAD phosphorylation can regulate 4EBP-1. Firstly, 4E-BP1 was identified through a Reverse Phase Protein Array (RPPA) antibody-based screen, which is a relatively unbiased method to differentiate the activities of key signaling molecule in multiple developmental pathways. 4E-BP1 was a top hit and to validate this, we analyzed mammary gland lysates, 3D mouse organoids and 3D MCF10A tubules and confirmed that BAD3SA-expressing cells were significantly decreased for phosphorylated 4E-

BP1. Finally, we confirmed that BAD3SA enhanced 4E-BP1 interactions with its binding partner eIF4E, thus providing a mechanism for diminished mRNA translation. Altogether, we provide convincing evidence for a functional association between BAD and 4E-BP1 regulation. We did not provide a mechanistic explanation for the BAD and 4E-BP1 regulation and submit that this is beyond the scope of this report (which is at the limit of total allowable figures and has an additional 9 Sup. Figs). We are currently exploring the mechanistic aspects of this model, which will form the basis of a follow-up report.

11. *Current findings of the manuscript suggest BAD protein may serve as a facilitator for the mammary gland development, not a regulator.*

Author response: We agree (see Schematic model, Sup. Fig. 9). “WT BAD is normally phosphorylated and permissive for pubertal gland morphogenesis”

General comments;

12. *Page 5 line 81-84. Why did the author choose BADS155A instead of BADS136A or BADS112A knock-in genetic mouse since the author showed significant changes in pBAD (ser112) and pBAD (Ser136) for WB and IF respectively in Figure 1? How different BAD phosphorylation including BADS155A, BADS136A and BADS112A influence on ductal elongation is unknown.*

Author response: Please see our response in (point 3 above) describing that all three serines are coordinately and temporally phosphorylated. Indeed, the *Bad*^{3SA} mouse phenocopies the *Bad*^{S155A} mouse, again supporting the concept of coordinated phosphorylation of all 3 serine residues. The *Bad*^{S136A} and *Bad*^{S112A} mouse has not been generated, while the *Bad*^{S155A} mouse has been well-characterized. Finally, we did not assess P-S155 because all commercial antibodies that we tested for P-S155 were non-specific and inappropriately showed positivity on the negative control *Bad*^{S155A} knock-in tissue.

13. *Page 6 line 113-114. In figure 3F, to prove the ectopically expressed human wild-type (WT) BAD or BAD3SA, it is necessary to detect Ser112,136 and 155 BAD phosphorylation protein.*

Author response: We have now included P-S136 of BAD alongside P-S112 of BAD in Fig. 3F. As mentioned above, we cannot confidently evaluate P-S155 as all commercially available P-S155 antibodies that we have tested show false positivity on *Bad*^{S155A} tissue.

14. *Page 7 line 117-118. Is there STR and mycoplasma free identification for MCF10A? I doubt there is something wrong with this cell line (contamination?).*

Author response: The MCF10A parental cell line was purchased from ATCC (see invoice below). We routinely test with a mycoplasma-specific PCR test (Eldering et al. 2004, *Biologicals: Journal of the International Association of Biological Standardization* 32: 183) and can demonstrate that the cells are free of mycoplasma.

CEDARLANE®

4410 Paletta Court
Burlington, Ontario L7L 5R2
Canada

(800) 268-5058 (North America)
Tel: (289) 288-0001 Fax: (289) 288-0070

www.cedarlanelabs.com

SOLD TO: UNIVERSITY OF ALBERTA
VENDU A: SEND INVOICE TO SHIP TO:

SHIP TO: [REDACTED]
EXPEDIE A: [REDACTED]

Page 1 of 1

Qty Ordered	Qty Shipped	Qty B/O	Catalog No. No. de Catalogue	Product Description/Description de Produit	
1	1	0	CRL-10317	MCF 10A; Breast, Human (Homo sapiens)	1 ml
LOT # 1					
Corporate Discount : Save 5%					
LIST BIO PRODUCT 100B(LB) DROP SHIPPED					

15. In general, blot quality is not adequate. Densitometry analysis is needed.

Author response: All blots were indeed quantitated and graphs are shown along representative blots in all figures. All blots were performed with at least 3 independent biological replicates. Bands were quantitated and statistical analysis was performed as indicated.

16. Page 9 line 159-163. PI3K-AKT-mTOR signaling is mainly affecting the phosphorylation of BAD at Ser136, but why didn't the author perform Western Blot on pBADSer136 instead of pBADSer112.

Author response: We have included western blots on pBADSer136 in Fig. 3f. Additionally, we have already shown phosphorylation of pBADSer136 in whole pubertal glands *in vivo*, and mouse organoids *ex vivo* (Fig. 1C and Sup Fig. 3a).

17. Page 13 line 248-249. mTOR can phosphorylate BAD at Ser136 by downstream substrate p70-S6K, it is necessary to study its expression to exclude the upstream signaling interference.

Author response: The expression of p70-S6K and phosphorylated p70-S6K is indeed shown in Fig. 5D-E, Sup. Fig 6G and Sup. Table 2.

Reviewer #3 (Remarks to the Author):

The manuscript by Goping and colleagues describes the exciting finding that Bcl-2 family member, BAD, may play a role in mammary gland development through the regulation of

localized translation by controlling the phosphorylation of the inhibitor of cap-dependent translation, 4E-BP1. Little is known about the connection of these proteins so the finding could open new doors in the study of 4E-BP1 biology and regulation of cap-dependent translation. Thus, based on its novelty and likely impact on the field, the manuscript is a good fit for publication in Nature Communications; however, I will raise a few issues and points for consideration that should be addressed prior to publication.

Author response: We thank the Reviewer!

1. *In Figure 1A, the authors provide expression data for pS112-BAD and total BAD via Western blot and its quantitation. Based on visual inspection of the blot, the quantitation does not appear to be correct. For example, the bands for total BAD in the puberty and pregnant samples look much more similar than the graphing would imply. Can the authors comment a bit more on how the quantitation was performed?*

Author response: The Reviewer is correct and the quantitation for this blot was not clarified. The lysates were derived from whole glands that are largely constituted of a fatty stromal compartment. Since BAD is expressed in the MECs, the epithelial marker, CK14, was used as a loading control (while p-BAD was quantitated to total BAD). Mammary gland lysates from pregnant animals have even lower epithelial cell content (CK14 levels), hence this may appear as a discrepancy when assessing the BAD immunoreactive band alone on western blots. We have clarified this in the text on line 616: “The proportion of epithelial to stromal cells is different between puberty/adult and pregnant/involuting glands as is indicated by cytokeratin 14 (CK14) intensity. Thus, CK14 is used as the loading control for the epithelial compartment.”

2. *In Figure 6A, the authors performed a cap pulldown assay to monitor eIF4E-4EBP1 binding. Because cap-dependent translation is initiated upon 4E-BP1 release and eIF4G binding to eIF4E through competition for the same binding site, the authors should also show data for eIF4G which should exhibit an opposite trend as 4E-BP1.*

Author response: This is a great suggestion and we have completed the experiment that does indeed show the expected trend. Please see lines 204 and Fig. 6a, with the text: “Hypophosphorylated 4E-BP1 inhibits translation by binding eIF4E and occluding the recruitment of eIF4G, which is normally required for subsequent eIF4F formation and translation initiation⁴⁶⁻⁴⁸. To examine whether 3SA disrupted normal eIF4E protein interactions, we used a m⁷GTP-cap pull-down assay to isolate cap-bound eIF4E. Indeed, we observed increased 4E-BP1 associated with eIF4E, with corresponding decreased association of eIF4G to eIF4E in 3SA-expressing cells compared to parental MCF10A and WT (Fig. 6a).”

3. *While the SUNSET assay demonstrates a measure of total translation, the authors imply that the effects are due specifically to regulation of cap-dependent translation. Assays should be performed to demonstrate this, such as the well-known luciferase-based cap-dependent/independent translation assay. It would also be nice to see data demonstrating specific effects on eIF4E protein-protein interactions in the protrusions using the PLA assay as has been published for eIF4E interactions with 4E-BP1 and eIF4G.*

Author response: Thank you for these suggestions. We now include the cap-dependent/independent translation assay with fluorescent markers to evaluate subcellular

localization. Please see line 234 and Sup. Fig. 7f): “Further, using a bicistronic fluorescent reporter assay, we observed that 3SA reduced cap-dependent and -independent translation (Sup. Fig. 7f)”. With respect to the suggestion of using the PLA assay to assess eIF4E interactions in the protrusions, we have instead included new data that addresses this same question, albeit indirectly. We quantified the levels of eIF4E, 4E-BP1 and P-4E-BP1 in the cyst vs the protrusions and found a significant decrease of hyperphosphorylated 4E-BP1 in the protrusions of 3SA cysts, while total 4E-BP1 levels were unchanged. See line 209: “Since 3SA induces protrusion-specific defects, we next assessed whether 4E-BP1 or eIF4E were differentially localized to protrusions. The body cells within the multicellular cysts and subcellular protrusions showed similar levels of total eIF4E and 4E-BP1 in both WT and 3SA (Fig. 6b; Sup. Fig. 7a). Hyperphosphorylated 4E-BP1 (p65_4E-BP1), however, was differentially expressed. It was similarly expressed in the body cells between the two genotypes, yet intriguingly, hyperphosphorylated 4E-BP1 was significantly reduced in 3SA protrusions (~2-fold decrease). Thus, 3SA protrusions are enriched for hypophosphorylated 4E-BP1 that is bound to eIF4E and prevents recruitment of eIF4G. Altogether these results suggest that 3SA inhibits mRNA translation locally within protrusions.”

4. As the authors indicated, additional kinases aside from mTORC1 have been demonstrated to phosphorylate 4E-BP1, namely CDK1 and CDK4. Have the authors explored any connections between these kinases and their findings regarding BAD? Perhaps some discussion of this could be included.

Author response: This is a great suggestion and we have included the text starting at line 291: “Thus, while the mTOR/4E-BP1 axis is well established, alternative 4E-BP1 kinases and phosphatases are known⁶⁰ and may be regulated by *Bad*^{3SA}. Candidate P-S65_4E-BP1 kinases include GSK3 β , ERK1/2, PIM2, p38MAPK, CDK1 and CDK2^{60,63,64}. While *Bad*^{3SA} mammary gland lysates showed no differences in phosphorylation of regulatory sites in 3 of these kinases (GSK3 β , CDK1 or ERK1/2; Sup. Table 2), the contribution of other kinases or phosphatases is unknown at this point.”

Reviewers' Comments:

Reviewer #1:

Remarks to the Author:

This is the second round of revisions for the manuscript entitled "BAD regulates mammary gland morphogenesis by 4E-BP1-mediated control and localized translation. The revised manuscript includes a few additional controls and new experiments, which moderately improved the overall strength of presented research. However, there are a series of unaddressed points that are crucial to support the author's conclusions.

The authors stated on their rebuttal that their results demonstrate a role for BAD on epithelial cells via the use of 3D cultures and mammary fatpad transplantation, which I agree. However, the authors have yet to demonstrate what is the real effect of BAD on epithelial cells. Is the pool of stem cells or progenitors reduced? Are luminal cells not fully differentiating thus the lack of branching? Is the overall epithelial lineage commitment altered? Again, these analyses are essential to support the authors conclusions, and to raise the motivation to study the role of BAD on mammary gland morphogenesis.

Also, the experiment testing MCF10A tubulogenesis assay presented on the rebuttal letter raises a series of concerns. Firstly, estrogen and progesterone are the classical pregnancy hormones, whereas prolactin is more of a late pregnancy/lactation hormone. So, the lack of phenotype could be because pregnancy signals were not really mimicked. Secondly, if pregnancy corrects the ductal elongation phenotype in BAD3SA mammary tissue, but pregnancy hormones do not correct tubulogenesis of MCF10A cells, this could indicate different regulatory mechanisms (cellular differentiation for example) that cannot be recapitulated with immortalized human cell lines, a point raised by other reviewers.

The experiments with MCF10A should be replaced with primary epithelial cells, given all the controls and for representing a best approach to address a developmental role for BAD.

Lastly, and for readability purposes, the pvalues for all experiments should be added to figures and figure legends.

Reviewer #2:

Remarks to the Author:

The authors specifically addressed all the comments raised by the reviewer. The present form of the revised manuscript is well-organized for publication, according to the journal's standard.

Reviewer #4:

Remarks to the Author:

BAD, a member of the proapoptotic Bcl-2 family proteins, is known for its role in apoptosis. In the present manuscript, the authors reported a non-canonical function of BAD in the pubertal mammary gland development. The observation that non-phosphorylated BAD represses localized translation required for focal adhesion maturation, cell protrusion stability, cell motility and mammary gland morphogenesis by inhibiting hyperphosphorylation of 4E-BP1 is intriguing. In the revised manuscript, the authors responded to most of the criticisms. The following comments should be addressed.

1. Some of the Western blot results and quantification could be confounded due to the poor quality of the images and choice of loading control. For example, in Figure 1, it is clear that pS112-BAD is increased during puberty, but the increase in total BAD in pregnant glands could be due to the low

abundance of CK14 in these glands.

2. Although the increase in pS136-BAD in TEB of pubertal mammary gland in Figure 1c is quite convincing, Fig. 3f showing the Western blot of S136-BAD is not impressive and the quantification does not match the visual inspection of the image.

3. It is hard to conclude anything from Figure 5d due to the poor quality of the image. For example, the background of S65-4E-BP1 is much darker in 5 wk Bad +/- compared to 4 wk Bad +/- lanes. The phosphorylation of 4E-BP1 may also cause mobility shift of total 4E-BP1.

4. Figure 6a: Based on the Western blot, it appears that 4E-BP1 and eIF4E binding is increased in cells expressing both WT and 3SA but eIF4G binding is increased in WT lysate but decreased in 3SA lysate.

5. One unanswered question is how non-phosphorylated BAD blocks hyperphosphorylation of 4E-BP1. In part of the Discussion, the authors implicated non-phosphorylated BAD to disrupt mTORC1 signaling (line 282-284 and 305-307). This is consistent with Figure 5e which shows that BAD3SA decreased both total and S2448-mTOR although it is not clear why decrease in mTOR is not associated with decrease in T389-p70 S6K. Thus, the authors also considered mTOR-independent mechanisms of 4E-BP1 regulation (line 290-300). However, in later part of the Discussion (line 329-331), the authors again implicated mTORC1 in 4E-BP1 hyperphosphorylation. A more consistent picture of 4E-BP1 regulation by BAD will be helpful.

6. Line 305-307. The reference should be included.

7. Line 353: 4EBP-1 should be 4E-BP1.

February 13, 2021

Point-by-Point response to Reviewers RE: NCOMMS-20-25093A-Z. BAD regulates mammary gland morphogenesis by 4E-BP1-mediated control of localized translation

Referee #1 (Remarks to the Author):

This is the second round of revisions for the manuscript entitled “BAD regulates mammary gland morphogenesis by 4E-BP1-mediated control and localized translation. The revised manuscript includes a few additional controls and new experiments, which moderately improved the overall strength of presented research. However, there are a series of unaddressed points that are crucial to support the author’s conclusions.

The authors stated on their rebuttal that their results demonstrate a role for BAD on epithelial cells via the use of 3D cultures and mammary fatpad transplantation, which I agree. However, the authors have yet to demonstrated what is the real effect of BAD on epithelial cells. Is the pool of stem cells or progenitors reduced? Are luminal cells not fully differentiating thus the lack of branching? Is the overall epithelial lineage commitment altered? Again, these analyses are essential to support the authors conclusions, and to raise the motivation to study the role of BAD on mammary gland morphogenesis.

Author response: Apologies that we did not completely address the original comment. Thank you for the clarification and we include the suggested experiments (Sup. Fig. 6). The corresponding text is in red and starts at line 166: “We next tested if BAD3SA altered epithelial cell lineage or the proportion of stem-like cells (Sup. Fig. 6). To examine epithelial cell lineage, primary mouse mammary epithelial cells were stained for the surface markers CD24 versus CD49f44. There was no significant difference in levels or proportion of luminal or basal epithelial subtypes. We next examined the epithelial stem/progenitor pools with the markers EpCAM versus CD49f to identify the Mammary Repopulating Unit (MRU) stem cells⁴⁵. MRUs can generate an entire functional mammary gland from a single cell⁴⁶. There was no difference in the MRU between the genotypes. Therefore, the Bad3SA defect likely manifested downstream of epithelial lineage commitment.” The methodology description starts at line 421.

Also, the experiment testing MCF10A tubulogenesis assay presented on the rebuttal letter raises a series of concerns. Firstly, estrogen and progesterone are the classical pregnancy hormones, whereas prolactin is more of a late pregnancy/lactation hormone. So, the lack of phenotype could be because pregnancy signals were not really mimicked. Secondly, if pregnancy corrects the ductal elongation phenotype In BAD3SA mammary tissue, but pregnancy hormones do not correct tubulogenesis of MCF10A cells, this could indicate different regulatory mechanisms (cellular differentiation for example) that cannot be recapitulated with immortalized human cell lines, a point raised by other reviewers.

Author response: If we understand correctly, you suggest that the 3SA ductal defect was rescued by pregnancy. If so, then pregnancy hormones should rescue the phenotype. We agree in principle, however, the major difference is that our data does not show that pregnancy corrects the phenotype. The phenotype recovers in nulliparous adult mice (Fig 2b-c). Since the mammary

gland has recovered in 8wk virgin animals, then as expected, mammary glands harvested from pregnant animals would also appear normal. We admit that our description of the developmental stages was not clear enough. To clarify this, we have added text on line 86 as “These results highlight that Bad3SA delays pubertal mammary gland development, which fully recovers by early adulthood”.

The experiments with MCF10A should be replaced with primary epithelial cells, given all the controls and for representing a best approach to address a developmental role for BAD.

Author response: We agree that data from primary epithelial cells should be included and this is done. Data from primary epithelial cells are shown in Fig 3a-d, Fig 5e, SFig 3a-c, SFig 6a-c, SFig 7e, and SFig 8c-d. Although these data were in the original submission, we had not explicitly labeled them as primary cells. This is now clearly stated as “primary mouse epithelial organoids” throughout the text. We respectfully retain the MCF10A experiments because (a) they confirm conservation between mouse and human cells, which is critical for future extrapolation to human conditions; and (b) they are genetically tractable and were useful for 4E-BP1 knock-down studies that demonstrated functional linkages.

Lastly, and for readability purposes, the pvalues for all experiments should be added to figures and figure legends. **Author response:** This is done.

Referee #2 (Remarks to the Author):

The authors specifically addressed all the comments raised by the reviewer. The present form of the revised manuscript is well-organized for publication, according to the journal's standard.

Author response: Thank you.

Referee #4 (Remarks to the Author):

BAD, a member of the proapoptotic Bcl-2 family proteins, is known for its role in apoptosis. In the present manuscript, the authors reported a non-canonical function of BAD in the pubertal mammary gland development. The observation that non-phosphorylated BAD represses localized translation required for focal adhesion maturation, cell protrusion stability, cell motility and mammary gland morphogenesis by inhibiting hyperphosphorylation of 4E-BP1 is intriguing. In the revised manuscript, the authors responded to most of the criticisms.

Author response: Thank you.

The following comments should be addressed.

1. Some of the Western blot results and quantification could be confounded due to the poor quality of the images and choice of loading control. For example, in Figure 1, it is clear that pS112-BAD is increased during puberty, but the increase in total BAD in pregnant glands could be due to the low abundance of CK14 in these glands.

Author response: We agree and have re-run lysates and improved Western blot quality. We also used a more consistent epithelial marker, CK18 (see Fig 1a).

2. Although the increase in pS136-BAD in TEB of pubertal mammary gland in Figure 1c is quite convincing, Fig. 3f showing the Western blot of S136-BAD is not impressive and the quantification does not match the visual inspection of the image.

Author response: We have re-run lysates and improved Western blot quality (Fig 3f).

3. *It is hard to conclude anything from Figure 5d due to the poor quality of the image. For example, the background of S65-4E-BP1 is much darker in 5 wk Bad +/- compared to 4 wk Bad +/- lanes. The phosphorylation of 4E-BP1 may also cause mobility shift of total 4E-BP1.*

Author response: We have improved Western blot quality (Fig 5d).

4. Figure 6a: Based on the Western blot, it appears that 4E-BP1 and eIF4E binding is increased in cells expressing both WT and 3SA but eIF4G binding is increased in WT lysate but decreased in 3SA lysate.

Author response: We have improved Western blot quality (Fig 6a).

5. *One unanswered question is how non-phosphorylated BAD blocks hyperphosphorylation of 4E-BP1. In part of the Discussion, the authors implicated non-phosphorylated BAD to disrupt mTORC1 signaling (line 282-284 and 305-307). This is consistent with Figure 5e which shows that BAD3SA decreased both total and S2448-mTOR although it is not clear why decrease in mTOR is not associated with decrease in T389-p70 S6K. Thus, the authors also considered mTOR-independent mechanisms of 4E-BP1 regulation (line 290-300). However, in later part of the Discussion (line 329-331), the authors again implicated mTORC1 in 4E-BP1 hyperphosphorylation. A more consistent picture of 4E-BP1 regulation by BAD will be helpful.*

Author response: We completely agree with this feedback. We have reorganized the discussion points. See lines 299-321 that are copied here:

- “We demonstrated that non-P-BAD inhibits pubertal development by interfering with mRNA translation. Translation is controlled by the kinase complex mTORC1 and mTORC1 loss-of-function induced similar transient pubertal mammary gland delay⁶². We therefore assessed whether Bad_{3SA} inhibits mTOR. Although Bad_{3SA} primary 3D organoids decreased both total and P_S2448 mTOR levels (Fig. 5e), these differences were not recapitulated in either whole mammary glands or MCF10A 3D tubules (Fig. 5d, Sup. Fig. 7g). Additionally, the mTORC1 downstream target p70-S6K was not differentially phosphorylated in any of the experimental models (Fig. 5d-e, Sup. Fig. 7g and Sup. Table 2), suggesting mTORC1 was not the target of the Bad_{3SA} defect. On the other hand, Bad_{3SA} consistently disrupted regulatory phosphorylation of the translational inhibitor, 4E-BP1. 4E-BP1 is also classically regulated by mTORC1 via sequential phosphorylation of T37, T46 and S65^{50, 66, 67}. Notably, phosphorylation of 4E-BP1 (T37/46) was not different between the genotypes, again, ruling out a direct role of mTORC1. Instead, Bad_{3SA} specifically blocked 4E-BP1 only at its hyperphosphorylation site (S65). Taken together, these results suggest that Bad_{3SA} regulated 4E-BP1 downstream of, or independent of, mTORC1. In a potentially similar scenario, RhoE regulates actin and focal adhesion assembly of NIH3T3 cells by inhibiting phosphorylation of 4E-BP1 on S65, independent of mTOR⁶⁸. Thus, while the mTOR/4E-BP1 axis is well established, alternative 4E-BP1 kinases and phosphatases are known⁶⁶ and may be regulated by Bad_{3SA}. Candidate P-S65_4E-BP1 kinases include GSK3 β , ERK1/2, PIM2, p38MAPK, CDK1 and CDK2^{66, 69, 70}. While Bad_{3SA} mammary gland lysates showed no differences in phosphorylation of regulatory sites in 3 of these kinases (GSK3 β , CDK1 or ERK1/2; Sup. Table 2), the contribution of other unexplored kinases

or phosphatases is unknown at this point. Indeed, this might explain why 4E-BP1 phosphorylation is unaffected by mTOR inhibitors in some cancer cells⁷¹⁻⁷³”

6. *Line 305-307. The reference should be included.*

Author response: Done

7. *Line 353: 4EBP-1 should be 4E-BP1.*

Author response: Done

Reviewers' Comments:

Reviewer #1:

Remarks to the Author:

The authors have addressed my concerns.

Please include a discussion regarding the meaning of data presented on Sup. Fig.6 in light of the overall phenotype noted in mice and 3D cultures

Reviewer #4:

Remarks to the Author:

The authors satisfactorily responded to the criticisms.

March 24, 2021

Point-by-Point response to Reviewers RE: NCOMMS-20-25093B. BAD regulates mammary gland morphogenesis by 4E-BP1-mediated control of localized translation

Referee #1 (Remarks to the Author):

The authors have addressed my concerns.

Author response: We thank the reviewer.

Please include a discussion regarding the meaning of data presented on Sup. Fig.6 in light of the overall phenotype noted in mice and 3D cultures

Author response: We have added the requested discussion. Starting at line 333: “Bad3SA disrupts pubertal mammary gland development and alters cell migration. While stem cells are critical for mammary gland morphogenesis, *Bad*^{3SA} does not appear to affect the stem/progenitor pools. *Bad*^{3SA} does not alter MRUs, which are capable of regenerating a functional mammary tree from a single stem cell⁴⁶. These MRUs serve as a source of differentiating luminal and myoepithelial cells and in line with this, the *Bad*^{3SA} mammary gland also has normal cell lineage proportions. Thus, *Bad*^{3SA} alters a morphogenetic process that is downstream of epithelial lineage commitment. Since epithelial cell motility is also critical for ductal elongation⁷⁶, *Bad*^{3SA}-mediated defects in cell motility are likely the cause of the developmental delay.”

Referee #4 (Remarks to the Author):

The authors satisfactorily responded to the criticisms.

Author response: We thank the reviewer.